# Sequential intrahost evolution and onward transmission of SARS-CoV-2 variants

Ana S. Gonzalez-Reiche [1], Hala Alshammary[2,3], Sarah Schaefer[4], Gopi Patel[4], Jose Polanco[2,3], Juan Manuel Carreño[2,3], Angela A. Amoako[2,3], Aria Rooker[2,3], Christian Cognigni[2,3], Daniel Floda[1], Adriana van de Guchte [1], Zain Khalil[1], Keith Farrugia[1], Nima Assad[5], Jian Zhang[1], Bremy Alburquerque [1,5], PARIS/PSP study group*, Levy A. Sominsky [2,3], Charles Gleason[2,3], Komal Srivastava[2,3], Robert Sebra [1,6,7,8], Juan David Ramirez[9,10], Radhika Banu[9], Paras Shrestha[9], Florian Krammer [2,3,9], Alberto Paniz-Mondolfi [9], Emilia Mia Sordillo [9] ✉, Viviana Simon [2,3,4,9,11] ✉ & Harm van Bakel [1,2,6] ✉

Persistent severe acute respiratory syndrome coronavirus 2 (SARS-CoV-2) infections have been reported in immune-compromised individuals and people undergoing immune-modulatory treatments. Although intrahost evolution has been documented, direct evidence of subsequent transmission and continued stepwise adaptation is lacking. Here we describe sequential persistent SARS-CoV-2 infections in three individuals that led to the emergence, forward transmission, and continued evolution of a new Omicron sublineage, BA.1.23, over an eight-month period. The initially transmitted BA.1.23 variant encoded seven additional amino acid substitutions within the spike protein (E96D, R346T, L455W, K458M, A484V, H681R, A688V), and displayed substantial resistance to neutralization by sera from boosted and/or Omicron BA.1-infected study participants. Subsequent continued BA.1.23 replication resulted in additional substitutions in the spike protein (S254F, N448S, F456L, M458K, F981L, S982L) as well as in five other virus proteins. Our findings demonstrate not only that the Omicron BA.1 lineage can diverge further from its already exceptionally mutated genome but also that patients with persistent infections can transmit these viral variants. Thus, there is, an urgent need to implement strategies to prevent prolonged SARS-CoV-2 replication and to limit the spread of newly emerging, neutralization-resistant variants in vulnerable patients.

The genomic landscape of severe acute respiratory syndrome coronavirus 2 (SARS-CoV-2) has been shaped by the emergence of antigenically diverse variants of concern (VOC)[1]. These viral variants carry mutations that render them more transmissible, more fusogenic, and/or permit escape not only from infection but also from vaccine-induced immunity[2–4]. Some variants such as the Omicron VOCs, which swept the globe starting November 2021, also display partial or complete resistance to therapeutic or prophylactic monoclonal treatments[5]. Over the past 14 months, Omicron sublineages with increasing numbers of mutations in spike have continued to emerge (e.g., BA.2, BA.4/5, XBB.1, XBB.1.5), posing great challenges to the containment of SARS-CoV-2 spread. This provided the rationale for formulation of bivalent SARS-CoV-2 booster vaccines that include an Omicron BA.1 or BA.5 spike in addition to the ancestral spike.

---

A full list of affiliations appears at the end of the paper.  *A full list of author affiliations appears at the end of the paper.
✉e-mail: Emilia.Sordillo@mountsinai.org; viviana.simon@mssm.edu; harm.vanbakel@mssm.edu

Most patients with coronavirus disease 2019 (COVID-19) clear the virus upon resolution of the acute infection. However, ongoing SARS-CoV-2 replication has been documented in a subset of immunocompromised individuals. In these chronically infected cases, recovery of infectious virus over several months and stepwise acquisition of mutations in spike has been observed, pointing to positive selection[6-13]. It has been speculated that prolonged intrahost viral evolution played a role in the emergence of several SARS-CoV-2 VOCs such as Alpha and Omicron[14,15], but clear evidence for forward transmissions of mutated variants from chronic infection cases has been lacking.

We describe herein a primary case of persistent SARS-CoV-2 Omicron BA.1 infection marked by intrahost evolution of a variant (Omicron BA.1.23) encoding seven additional amino acid substitutions in the already antigenically distinct Omicron BA.1 spike protein, and that resulted in at least five further cases of Omicron BA.1.23 infection. Persistent infections documented in two of these cases (one lasting four weeks, the other more than four months) were associated with continued BA.1.23 evolution and acquisition of additional mutations within and outside of spike. Although the variants that evolved in the persistently infected cases during the cumulative eight-month period reveal unique constellations of mutations in each, they also strongly point to convergent viral evolution with other co-circulating SARS-CoV-2 lineages. Notably, most amino acid changes occurred at positions known to confer either immune escape[16,17] or altered viral fusogenicity[18-20]. Some of the escape mutations in BA.1.23 have also emerged in later Omicron lineages such as BA.2.75.2. Overall, our results indicate that persistent viral replication in the context of sub-optimal immune responses is an important driver of SARS-CoV-2 diversification.

## Results

### Emergence of a novel BA.1 sublineage through intrahost evolution

We performed genomic analysis of serially collected nasopharyngeal (NP) and anterior nares (AN) samples from an immunocompromised patient (P1) with diffuse B-cell lymphoma and persistent SARS-CoV-2 Omicron BA.1 replication between December 2021 and March 2022. Over a 12-week period, we documented the accumulation of nine

amino acid substitutions in the spike protein N-terminal domain (NTD), the receptor binding domain (RBD), and in the S1/S2 furin cleavage site (FCS) (Fig. 1) within the same patient. The first four mutations R346T, K458M, E484V, and A688V were detected simultaneously 40 days after the initial SARS-CoV-2 diagnosis and were fixed. Two weeks later (day 64), L167T and the FCS mutation P681Y were detected in addition to the four initial mutations. During the following weeks additional mutations emerged; samples from this period contained shared (L455W) as well as distinct signature mutations (E96D on day 72, S477D on day 81). Notably, only two mutations emerged outside of the spike gene (Supplementary Fig. 1, P1), suggesting positive intrahost selection of spike protein changes due to competitive replication advantages. The SARS-CoV-2 substitution rate varied throughout the course of infection; after an initial period of three weeks without changes in the consensus sequence, there was a rapid accumulation of substitutions between week 4 and week 12. On average, one substitution per week was observed during this period, corresponding to a rate of 52 substitutions/year – approximately two-fold higher than the global average of 26–27 substitutions/year[21,22].

### Forward spread confirms transmission potential of the novel BA.1.23 sublineage

Background health system-wide SARS-CoV-2 genomic surveillance conducted by the Mount Sinai Pathogen Surveillance Program (MS-PSP) during the same time period identified three other patients harboring SARS-CoV-2 Omicron variants that shared the same combination of spike amino acid substitutions found in the index case P1 (E96D, R346T, L455W, K458M, E484V, H681R, A688V), as well as the synonymous mutation T6001C. A query of more than 13.8 million global SARS-CoV-2 sequences deposited in the GISAID (Global Initiative on Sharing Avian Influenza Data) database up to November 2022 revealed only two additional related genomes, both originating from the NYC area (Fig. 2, source data are provided as a Source Data file). Based on the metadata provided, these two genomes were obtained from distinct individuals that differed by age and gender from our patients. Altogether, the presence of the same unique combination of mutations in five additional cases indicates limited local forward transmission of this novel Omicron subvariant, which received the BA.1.23 Pango lineage designation (Fig. 2, Table 1).

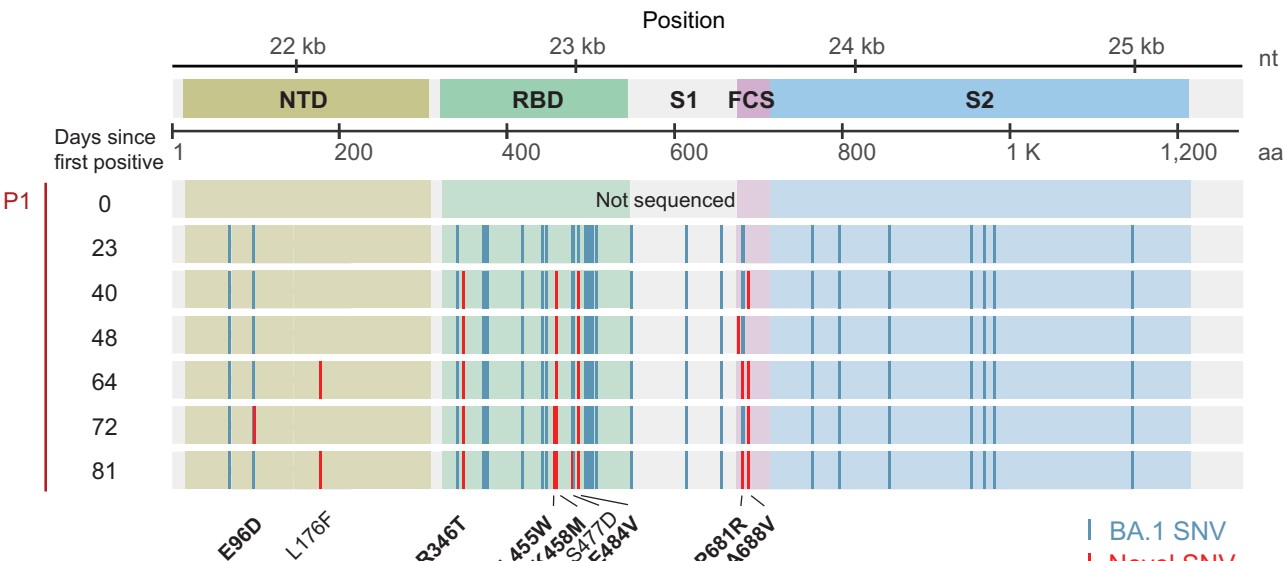

**Fig. 1 | Intrahost emergence of an Omicron BA.1.23 subvariant.** Multiple sequence alignment of the SARS-CoV-2 spike gene indicating the appearance of single nucleotide variants (SNVs) in the consensus sequence of the spike gene in sequential specimens obtained from the index case (Patient 1, P1). Novel SNVs relative to the ancestral BA.1 strain are shown in red and labeled at the bottom of the figure. Bold labels denote signature mutations observed in multiple specimens. Consensus changes in BA.1 compared to the Wuhan-1 strain are shown in blue.

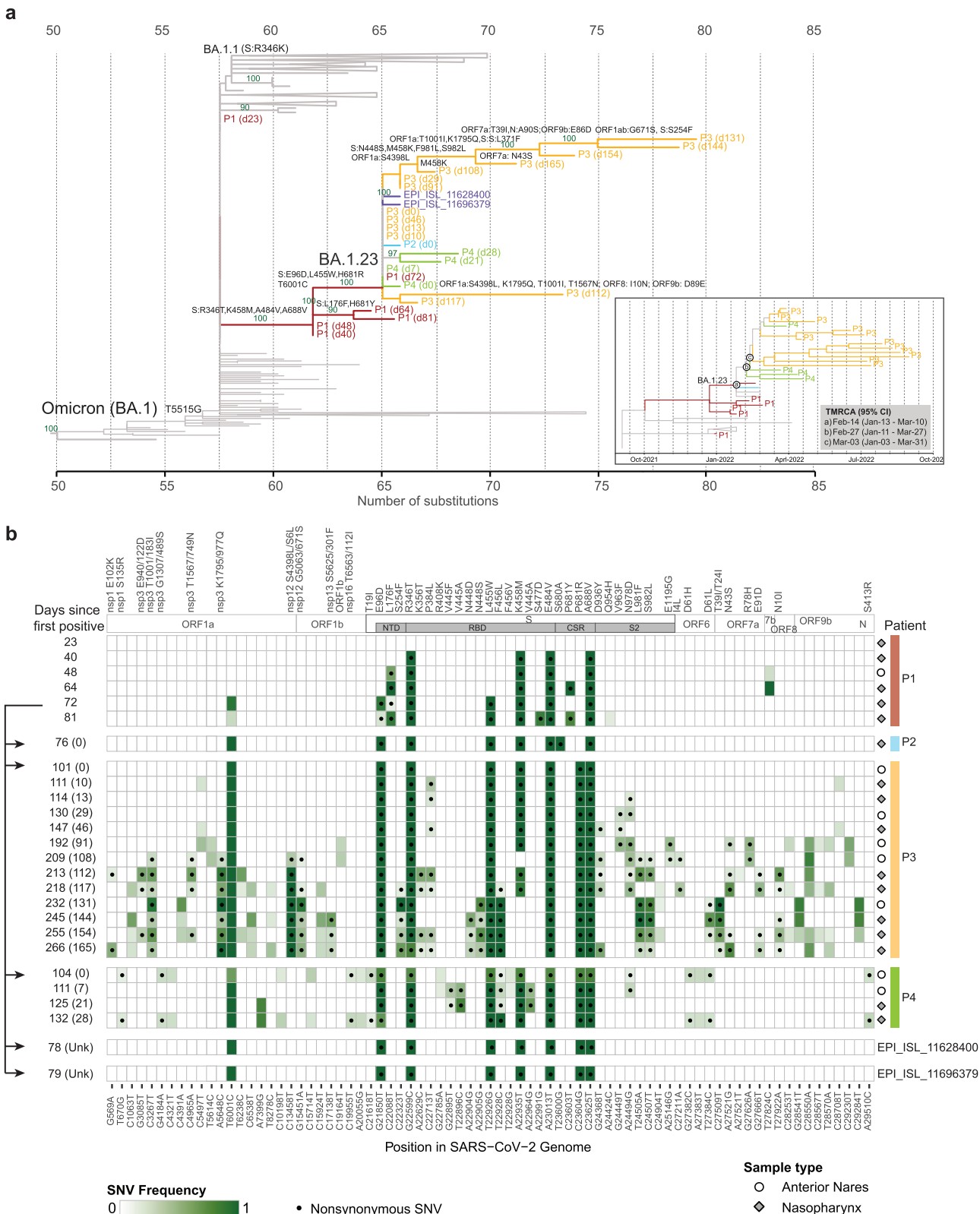

All transmission cases contained one synonymous change in ORF1a (T6001C) previously only observed at day 72 in the index case (Supplementary Fig. 1). This narrowed the time window of the first transmission to an 18-day period following day 64, two months after the first positive SARS-CoV-2 test of patient P1. During this time, patients P2 and P4 were admitted to the same unit as P1 for 10 and 18 days, respectively. Following a transfer, P4 also overlapped for two

days with P3 in a different unit. While this provided a potential opportunity for direct or indirect contact, it is important to note that P3 and P4 were not tested for SARS-CoV-2 until three weeks later, when they were found to be positive. No contact information was available for the two cases identified outside our health system. As of February 2023, the most recent case was detected in mid-April 2022 and the last SARS-CoV-2 positive specimen from a forward transmission case (P3)

**Fig. 2 | Forward transmission of the Omicron BA.1.23 subvariant. a** Maximum likelihood (ML) phylogenetic subtree with SARS-CoV-2 (BA.1) sequences from the persistent infection case P1 (red) and the onward transmissions (P2, P3, P4 in cyan, yellow and green respectively), in a global background of sequences available in GISAID. Branches are colored to identify each patient. The number of days after the first SARS-CoV-2 positive specimen of P1 is indicated in brackets. Sibling clusters are collapsed for easier visualization. The x-axis shows the number of nucleotide substitutions relative to the root of the phylogenetic tree. Bootstrap support values above 70% are shown for the un-collapsed branches. The distinct BA.1 subvariant that was transmitted was designated as PANGO lineage BA.1.23. The bottom-right insert shows a time-scaled ML phylogeny and the estimates for the time of the most recent ancestors (TMRCA) and 95% confidence intervals, for the nodes where the transmissions from the index case are positioned. **b** Frequency of single nucleotide variants (SNV), and amino acid substitutions for SARS-CoV-2 genomic positions with consensus changes from ancestral Wuhan-1 and Omicron BA.1. Positions with

mixed nucleotides below consensus levels are also shown for intrahost SNVs (miSNVs) seen in more than one time point. The sequenced specimens are shown sequentially for P1 with prolonged BA.1 infection and transmission cases of BA.1.23 (P2, P3 and P4). The viral genomes from P1 show progressive accumulation of mutations in the N-terminal domain (NTD), receptor binding domain (RBD), and S1/S2 furin cleavage site (FCS); The same constellation of mutations was subsequently detected in three documented transmission cases (P2, P3 and P4). The number of days since the first positive test in P1 is shown on the left, with the number of days after the first positive test for each patient between brackets. Positions with non-synonymous SNVs are marked by filled circles. Specimen type is indicated on the right by open circles (anterior nares) or filled diamonds (nasopharynx). Positions are numbered according to the reference genome sequence NC_045512.2. Arrows on the left indicate the likely window of transmission from the index patient between days 64–72. Source data is provided in the Source Data file.

was collected in September 2022. Although viral isolation failed for specimens available from P1, we successfully cultured SARS-CoV-2 from the initial positive specimens of the subsequent cases (P2–P4), confirming the transmission of replication-competent virus.

## Sequential persistent infections drive further intrahost evolution of BA.1.23

All three forward transmissions in our health system were detected in patients with underlying hematologic malignancies. Although patient P2 cleared the BA.1.23 infection, patients P3 and P4 both developed persistent infections (Figs. 2 and 3a). The MS-PSP continued monitoring these patients during follow-up visits and hospitalizations. Patient P4 remained positive by nucleic acid amplification tests (NAAT) for four weeks, with acquisition of an additional mutation in the spike RBD (S:V445A) within one week of the initial positive test (Figs. 2 and 3a). Patient P3 developed a much longer persistent infection lasting for more than four months. Genomic analysis of all 13 serially collected specimens from patient P3 identified several new amino acid substitutions throughout the viral genome (Fig. 2b and Supplementary Fig. 1). The most divergent (i.e. highest number of mutations) specimen collected 131 days after patient P3's COVID-19 onset contained 11 additional amino acid substitutions compared to the originally transmitted BA.1.23 variant. These included six substitutions in the spike NTD (S254F), RBD (N448S, F456L, reversion of 458 M to K), and S2 (reversion of 981 L to F, S982L); as well as five substitutions in other SARS-CoV-2 proteins (ORF1a nsp3: T1001/183I, K1795/977Q, ORF1ab nsp12: S4398/6 L, G5063/671 S; and ORF7a: T39I/T24I). Notably, two spike amino acid reversions to the Wuhan-1 (S:M458K) or BA.1 (S:F981L) sequence were accompanied by changes at neighboring positions (S:F456L and S:S982L) (Fig. 2b and Supplementary Fig. 1). Interestingly, in the subsequent three swab specimens (days 144, 154 and 165) the number of amino acid substitutions decreased relative to the specimen collected on day 131, with recurrent flips and reversions across specimens, suggesting the presence of competing quasispecies.

To further investigate this, we looked for positions with intrahost single nucleotide variants (iSNVs) that were present in only a minority of the SARS-CoV-2 viral population within each specimen (referred to as miSNVs). We identified miSNVs within or outside the spike gene in each of the three persistent infection cases (P1, P3, and P4) with several instances in which miSNV were present in earlier specimens, prior to their fixation in subsequent specimens from the same patient (Fig. 2b and Supplementary Figs. 1, 2). We also observed numerous positions with miSNVs that persisted across multiple sequential specimens without ever becoming dominant. This was most notable for patient P3, where the number of positions with miSNVs increased from 0 to 32 between days 101 and 266 – outnumbering the consensus sequence changes by a factor of three (Fig. 2b and Supplementary Fig. 3). Approximately half of these mutations occurred within the spike gene, where nonsynonymous miSNVs were clustered in the S1 (NTD:S254F,

RBD:K356T, S371F, P384L, F456L, K458M, and FCS:N679K) and S2 (D936Y, V963F, N978D, L981F, S982L, E1195G) domains (Fig. 2b and Supplementary Fig. 4). Mutations at these positions are rare at the consensus level, with a maximum prevalence of 0.1% in the GISAID database [as of 2022-09-10]. Although we found no clear evidence of compartmentalization of miSNVs by specimen source across patients, there was a notable decrease in miSNVs and increase in consensus mutations in the AN specimen collected from patient P3 on day 131, compared to the earlier and later NP specimens. Thus, intrahost adaptation occurs with different dynamics pointing to bottlenecks and competing selection pressures, resulting in the appearance and disappearance of specific mutations.

To examine a potential role for recombination in the diversification of BA.1.23 we used RIPPLES[23] to assess the phylogenetic placement of genome segments from the most divergent consensus genotypes of BA.1.23 within the global SARS-CoV-2 diversity. We further queried if the observed miSNV patterns were present in contemporary lineages using covSPECTRUM[24] and considering all patterns in a sliding window of 3 consecutive miSNVs. Neither analysis yielded evidence for recombination, providing further support for diversification through the accumulation of mutations during intrahost evolution.

## Impact of treatment and host immunity on BA.1.23 evolution

To determine the potential impact of SARS-CoV-2 antiviral treatments on intrahost evolution we examined the medication histories of patients P1 (index), P2, P3 and P4 (Fig. 3, source data are provided as a Source Data file). We also assessed SARS-CoV-2 spike binding antibody levels using available serological data from clinical tests. The index patient P1 was vaccinated at the time of admission with two doses of an unspecified vaccine administered six months and one week before hospitalization (Fig. 3c). A third vaccine dose was administered two days after admission. Additional SARS-CoV-2 antiviral treatments included a course of remdesivir on days 1-6 and a dose of Gamunex IgG on day 9 (Fig. 3D). Moderate titers of SARS-CoV-2 spike binding antibodies were detected on day 38, around the time the first intrahost mutations were found (Figs. 2b and 3b). These antibody titers could be due to residual antibodies from the Gamunex IgG treatment, an immune response to vaccination and/or infection, or a combination of treatment and host response. Notably, the detection of the first three spike mutations followed a rapid increase in virus detected in nasal secretion (NAAT mean cycle thresholds (Ct) decreased from 35 on day 30 to 28 on day 40), suggesting a potential selection bottleneck around this time (Figs. 3a and 3c).

Patient P2 received four doses of Moderna mRNA vaccine, three of them at least four months prior to their first positive SARS-CoV-2 test and had high titers of SARS-CoV-2 antibodies when assayed three months before their single positive SARS-CoV-2 NAAT (Fig. 3c). This patient also received a one-month course of nirmatrelvir/ritonavir (Paxlovid™) (Fig. 3d). The combination of high levels of spike binding

**Table 1 | Spike mutations detected in the sequenced cases**

Mutations in Spike relative to earliest sample (BA.1: A67V, T95I, Y145D, L212I, G339D, S371L, S373P, S375F, K417N, N440K, G446S, S477N, T478K, E484A, Q493R, G496S, Q498R, N501Y, Y505H, T547K, D614G, H655Y, N679K, P681H, N764K, D796Y, N856K, Q954H, N969K, L981F)

| Patient | Sex | Age range | Days from P1 first positive specimen | GenBank Accession number | Source | Lineage | E96 | L176 | S254 | R346 | S371 | V445 | N448 | L455 | F456 | E484 | T547 | S680 | P681 | A688 | L981 | S982 |
|---|---|---|---|---|---|---|---|---|---|---|---|---|---|---|---|---|---|---|---|---|---|---|
| P1 | M | 61–65 | 0 | N/A | This study | BA.1 | | | | | | | | | | | | | | | | |
| | | | 23 | ON220548 | This study | BA.1 | | | | | | | | | | A | | | H | V | F | |
| | | | 40 | ON220539 | This study | BA.1 | | | | T | | | | | M | V | | | H | V | F | |
| | | | 48 | ON196014 | This study | BA.1.23 | | | | T | | | | | M | V | | | H | V | F | |
| | | | 64 | ON193425 | This study | BA.1.23 | | F | | T | | | | | M | V | | | Y | V | F | |
| | | | 72 | ON220529 | This study | BA.1.23 | D | | | T | | | W | | M | V | | | H | V | F | |
| | | | 81 | ON220571 | This study | BA.1.23+ | D | F | | T | | | W | | M | V | D | | Y | V | F | |
| P2 | F | 66–70 | 76 | ON220533 | This study | BA.1.23 | D | | | T | | | W | | M | V | | A | H | V | F | |
| P3 | M | 36–40 | 101 | ON934538 | This study | BA.1.23 | D | | | T | | | W | | M | V | | | R | V | F | |
| | | | 111 | ON934577 | This study | BA.1.23 | D | | | T | | | W | | M | V | | | R | V | F | |
| | | | 114 | ON934030 | This study | BA.1.23 | D | | | T | | | W | | M | V | | | R | V | F | |
| | | | 130 | ON934607 | This study | BA.1.23+ | D | | | T | **F** | | W | | M | V | | | R | V | F | |
| | | | 147 | n.s.* | This study | BA.1.23 | D | | | T | | | W | | M | V | | | R | V | F | |
| | | | 192 | n.s.* | This study | BA.1.23+ | D | | | T | **F** | | W | | M | V | | | R | V | F | |
| | | | 209 | n.s.* | This study | BA.1.23+ | D | | | T | **F** | | W | | M | V | | | R | V | F | |
| | | | 213 | n.s.* | This study | BA.1.23 | D | | | T | | | W | | M | V | | | R | V | F | |
| | | | 218 | n.s.* | This study | BA.1.23+ | D | | | T | **F** | | W | | M | V | | | R | V | F | |
| | | | 232 | n.s.* | This study | BA.1.23+ | D | | F | T | **F** | | S | L | | V | | | **R** | V | F | **L** |
| | | | 245 | n.s.* | This study | BA.1.23+ | D | | | T | **F** | | D | L | | V | V | | R | V | F | **L** |
| | | | 255 | n.s.* | This study | BA.1.23+ | D | | | T | **F** | | S | L | | N | V | | R | V | F | **L** |
| | | | 266 | n.s.* | This study | BA.1.23+ | D | | | T | **F** | | W | L | | N | V | | R | V | F | |
| P4 | F | 61–65 | 104 | n.s.* | This study | BA.1.23+ | D | | | T | **F** | | W | | M | **V** | K | | **R** | V | F | |
| | | | 111 | ON854465 | This study | BA.1.23 | D | | | T | | | W | | M | **V** | | | **R** | V | F | |
| | | | 125 | ON619382 | This study | BA.1.23+ | D | | | T | | A | W | | M | **V** | | | **R** | V | F | |
| | | | 132 | n.s.* | This study | BA.1.23+ | D | | | T | | | W | L | M | **V** | | | **R** | V | F | |
| S1 | F | 81–85 | 78 | ON118042 | Howard et al. 2022. GenBank | BA.1.23 | D | | | T | | | W | | M | **V** | | | **R** | V | F | |
| S2 | F | 26–30 | 79 | ON139241 | Howard et al. 2022. GenBank | BA.1.23 | D | | | T | | | W | | M | **V** | | | **R** | V | | |

*n.s. Consensus not submitted due to presence of >10 intrahost single nucleotide variants. N/A: not sequenced.

*Mutations in bold correspond to changes at positions already mutated in BA.1.

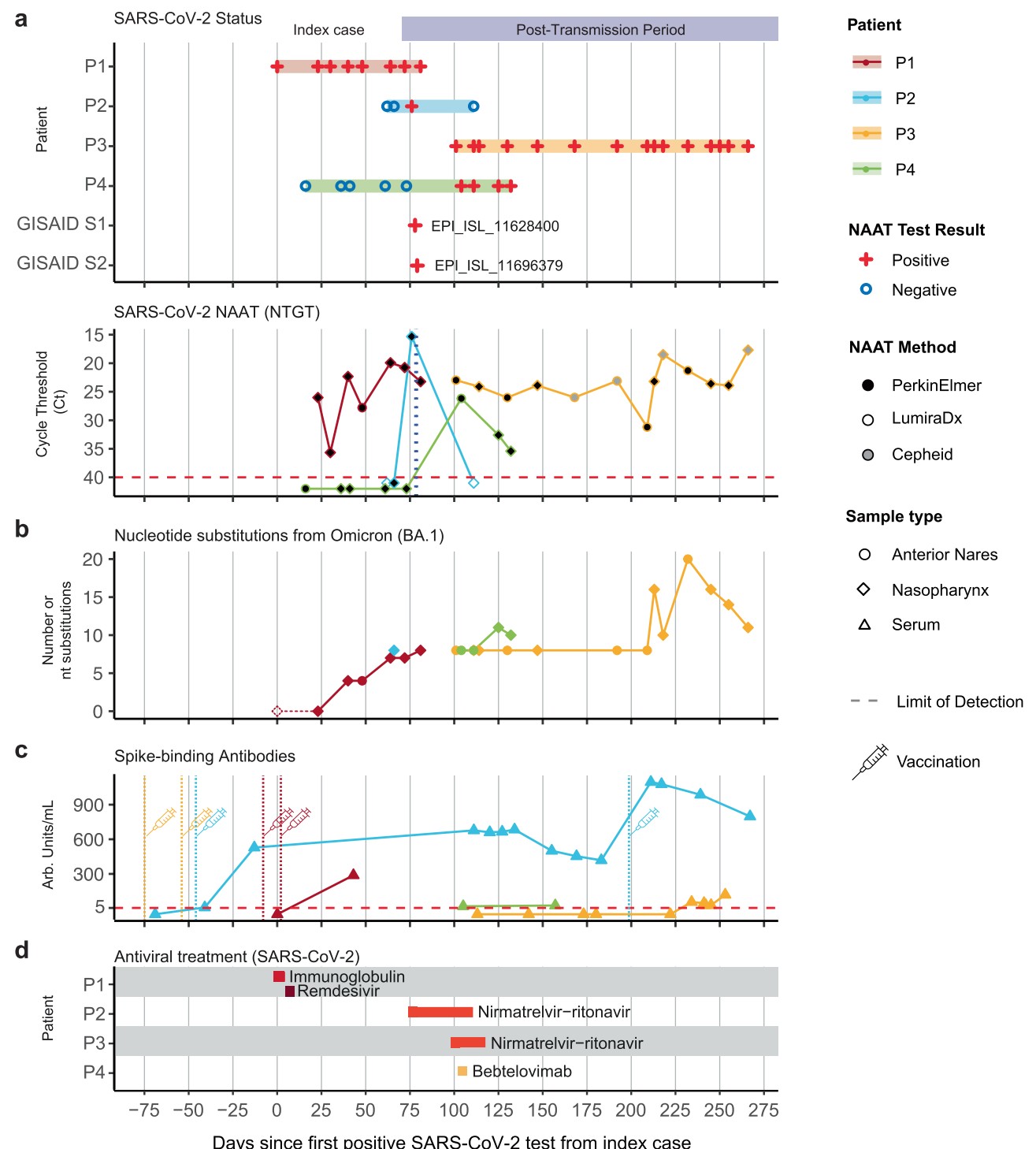

**Fig. 3 | Timeline of BA.1.23 evolution compared to antibody levels and treatments.** a The timeline of BA.1.23 infected patients' positive (red crosses) or negative (open blue circles) nucleic acid amplification test (NAAT) for SARS-CoV-2 (top). The N gene target cycle threshold (Ct) values for respiratory specimens, for different diagnostic methods is shown (bottom panel). The NAAT panel only includes data points from positive tests that reported a Ct value. b Number of nucleotide substitutions in the consensus sequence relative to Omicron BA.1 in the sequenced specimens. c SARS-CoV-2 spike-binding IgG antibody levels for P1–P4. Antibody levels are shown in arbitrary units per mL (Arb. units/mL). Documented vaccine administrations are indicated. d Timeline of SARS-CoV-2 antiviral treatments received by P1–P4. Treatment duration is indicated by the length of the bar. Source data provided in the Source Data file.

IgG antibodies and antiviral treatment may explain why this patient successfully cleared the BA.1.23 infection.

Patient P3 was also vaccinated with two doses of Pfizer mRNA vaccine at least five months prior to their first positive test and received a three-week course of nirmatrelvir/ritonavir (Paxlovid™). However, no SARS-CoV-2 spike binding IgG antibodies were detected during the first four months of the persistent infection (Fig. 3c). Notably, during this time we also did not detect new substitutions in the BA.1.23 virus, despite indications of active viral replication based on consistently low NAAT Ct values (less than 25) in NP and AN samples (Fig. 3a). Active BA.1.23 replication was confirmed by isolation of replication-competent SARS-CoV-2 from five specimens collected

between study days 101–147. The appearance of new intrahost substitutions during the last month of persistent infection occurred around the time of detection of SARS-CoV-2 antibodies at levels close to the limit of detection of the serological assay used (COVID-*SeroKlir*, *Kantaro* Semi-Quantitative SARS-CoV-2 IgG; Fig. 3). Since patient P3 did not receive any biologicals during their hospitalization, it is likely that the observed seroconversion was the result of a delayed and weak host immune response.

For patient P4 no prior vaccinations were registered. Bebtelovimab was administered after their first positive SARS-CoV-2 PCR test, followed by a course of nirmatrelvir/ritonavir (Paxlovid™). P4 had low SARS-CoV-2 antibody titers at the time of the first positive PCR test prior to the monoclonal antibody treatment (Fig. 3c). Of note, a single S:V445A mutation emerged in this patient. Viral variants carrying this mutation have been associated with reduced susceptibility to bebtelovimab in pseudotyped virus-like particle (VLP) neutralization assays[25]. We did not find SARS-CoV-2 mutations associated with remdesivir[26–31] or nirmatrelvir/ritonavir resistance[32] in any of the patients treated with these drugs.

### Serum neutralization profile of the transmitted persistent Omicron BA.1.23 variant

Neutralization of Omicron BA.1 isolates by sera from convalescent or vaccinated individuals is strongly reduced compared to the levels obtained for neutralization of the ancestral strains[33–35]. Therefore, we assessed the degree of neutralizing resistance of the transmitted BA.1.23 variant (BA.1 + S:E96D, R346T, L455W, K548M, E484V, P681R, A688V) compared to Omicron BA.1 (B.1.1.529), as well as the ancestral SARS-CoV-2 (USA-WA1/2020, WA). We used a multi-cycle micro-neutralization assay in which human serum is present continuously to best mimic physiological in vivo conditions[33,36]. We selected sera from two subsets of PARIS study participants representing distinct levels of immunity. The first panel of sera tested was collected before and after booster vaccination, while the second set of samples was obtained before and after Omicron BA.1 breakthrough infection in vaccinated participants (see Supplemental Table 1 for details).

Sera collected prior to booster vaccination with monovalent SARS-CoV-2 RNA vaccines (18 matched samples), neutralized BA.1 and BA.1.23 less well than WA1/USA (geometric mean titer; GMT WA1: 71; GMT BA.1: 12; GMT BA.1.23: 6; Fig. 4a, source data are provided as a Source Data file), with the majority of samples failing to display any neutralization activity against BA.1 (5/9) and BA.1.23 (7/9), respectively. Matched sera collected after mRNA booster vaccination from the same study participants showed the booster vaccination increased the neutralization titers for all the viral isolates, but the loss in neutralization for BA.1.23 was greater than 12-fold compared to the >6-fold reduction for BA.1 relative to WA1/USA (geometric mean titer; GMT WA1: 1,689; BA.1: 189; GMT BA.1.23: 43; Fig. 4a). Of note, the loss of neutralization activity for BA.1 relative to the WA1/USA ancestral variant measured here is less than what we and others have previously reported which could be due to assay variation in combination with the limited number of samples tested[33,34,37].

We next tested how sera collected from the same study participants before and after breakthrough infection with the BA.1 Omicron variant (22 matched samples) would neutralize BA.1.23. We noted a significant difference in neutralization activity for BA.1.23 and BA.1 compared to WA1 in the samples collected after BA.1 breakthrough infection (Fig. 4b). Prior to BA.1 infection, the difference was >17-fold with 3/11 samples having undetectable neutralization activity for BA.1.23 and >14-fold difference with 2/11 of the samples failing to neutralize BA.1 (GMT WA1: 346; GMT BA.1: 26; GMT BA.1.23: 22; Fig. 4b). After infection, neutralization titers increased for all three viruses reducing the difference to four-fold for BA.1 and five-fold for BA.1.23 (GMT WA1: 1,913; GMT BA.1: 503; GMT BA.1.23: 399; Fig. 4b).

Then, we analyzed sera collected from patient P2 at different time points (i.e., before booster vaccination, after booster vaccination but before BA.1.23 infection, and at two time points after infection with BA.1.23). P2 mounted neutralizing antibodies after booster mRNA vaccination against WA1, but not against BA.1 or BA.1.23 (Fig. 4c). Serum collected one month after the breakthrough infection with BA.1.23 showed a sharp increase in neutralization activity for BA.1 and BA.1.23. (Fig. 4c). Lastly, we computed the fold changes for each viral isolate before and after booster vaccination or breakthrough infection with either BA.1 or BA.1.23 (Fig. 4d). These data show that monovalent booster vaccination increases neutralization titers for the ancestral variants, while breakthrough infections with Omicron variants yielded a boost in neutralization of these contemporary variants. Interestingly, we observed large variation in the extent to which BA.1 infection boosted neutralization of the two different Omicron variants (Fig. 4d, middle panel, fold changes range from less than 1 to more than 100-fold difference).

In conclusion, the transmitted persistent BA.1.23 isolate is more neutralization-resistant than the parental BA.1. However, infection with either Omicron variant BA.1 or BA.1.23 induced cross-reactive humoral responses that were capable of neutralizing both variants.

## Discussion

It has been speculated that the emergence of antigenically diverse SARS-CoV-2 variants such as Omicron result from intrahost viral evolution driven initially by suboptimal immune responses and then spread by forward transmissions[10]. Our data demonstrate that intrahost evolution of SARS-CoV-2 during persistent infection in immunocompromised individuals can drive the emergence and spread of novel (sub)variants with extensive mutations in key antigenic regions, even in the context of the already highly mutated Omicron lineage. Notably, we find that transmission of persistent viruses can occur as late as two months after the first positive SARS-CoV-2 test (Fig. 2b). Further sequential evolution in transmission cases shows that rapid divergence can occur stepwise and persist in small numbers of individuals, complicating early detection of novel variants.

The most extensive changes in the evolution of the BA.1.23 lineage were seen during its emergence in patient P1 and its subsequent divergence in patient P3 over a total period of 8 months. Based on available clinical testing data, both patients were initially negative for SARS-CoV-2 antibodies but seroconverted to moderate-to-low levels around the time spike mutations were accumulating. These conditions are likely optimal for the selection of immune escape mutations and provide an explanation for the clustering of BA.1.23-specific changes in the spike protein. It is important to note that the antiviral treatments administered over limited time periods as monotherapies (e.g., monoclonal antibodies, remdesivir, and nirmatrelvir/ritonavir (Paxlovid™)) did not eliminate the persistent infection, highlighting the need for improved or potential combination therapy tailored for viral clearance in these special situations.

During the initial emergence of the BA.1.23 lineage in index patient P1, mutations were almost exclusively non-synonymous and concentrated in the spike protein. The amino acid substitutions or positions at which changes accumulated have been associated with neutralization escape (e.g., R346T, E484V, S477)[16,17], increased angiotensin converting enzyme 2 (ACE2) binding avidity (e.g., R346T, L455W)[38], improved viral fusogenicity (e.g., P681R/Y)[18], or a predicted increase in fitness (e.g., positions 346, 484 and 681)[39]. Subsequent stepwise changes in patient P3 occurred more broadly throughout the SARS-CoV-2 genome. The RBD mutations N448S and F456L present in the most recent specimen from P3 are within the ACE2 binding region with the potential for changes in binding affinity and antibody escape[38]. Mutations outside the spike mapped to nonstructural proteins, many of which have previously been detected in other VOCs

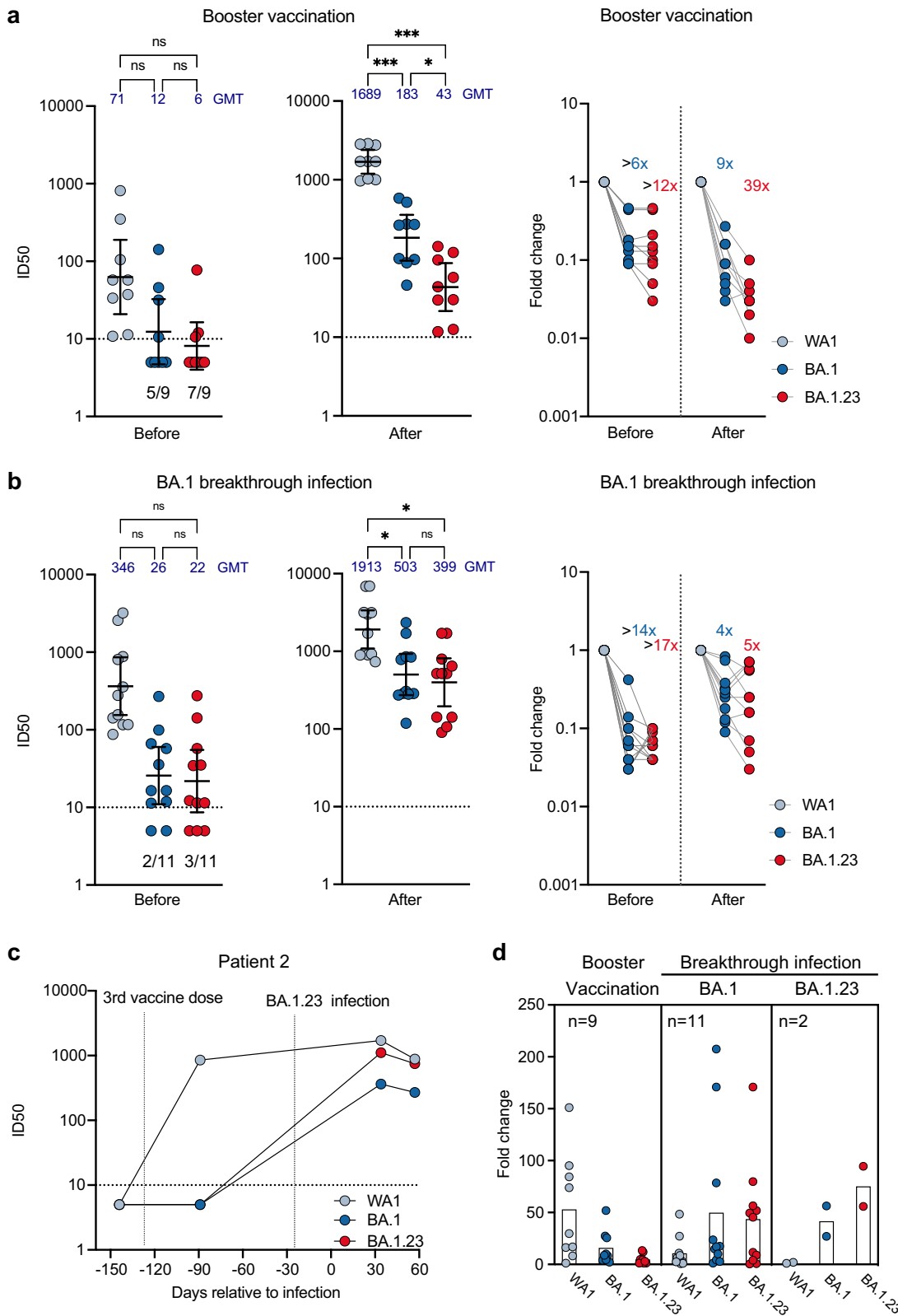

such as Nsp3:T183I (Alpha), Nsp3:K977Q (Gamma), Nsp12:G671S (Delta AY.*) and Omicron (BA.2.75). Although the effects of these mutations have not been characterized, they occurred in the virus polymerase subunit (Nsp12) and in proteins with antagonistic activity towards the host's innate immunity (Nsp3, ORF6, ORF7a, ORF9b)[40].

Consistent with the extensive changes in the spike protein, the resistance of the transmitted BA.1.23 variant to neutralization by polyclonal sera was substantially increased compared to the ancestral SARS-CoV-2 strain as well as to BA.1 Omicron. This was true both for sera collected before and after monovalent booster vaccination, as well as BA.1 breakthrough infection (Fig. 4). Escape from neutralization was likely mediated by the combination of R346T, L455W, K458M, and E/A484V within the receptor binding region of spike. Of note, substitutions at position R346 are found in several Omicron sublineages

**Fig. 4 | The Omicron BA.1.23 subvariant displays strongly reduced neutralizing activity compared to the parental Omicron variant as well as ancestral SARS-CoV-2 variant. a** Absolute neutralization titers (left) and fold-reduction (right) for WA−1, BA.1 and BA.1.23 variants by paired sera from 9 study participants collected before and after booster vaccination ($n = 18$). The number of samples with titers below the limit of detection of the serological assay (dashed line) is indicated at the bottom of the graph. The error bars represent the geometric mean with 95% confidence intervals. A one-way RM ANOVA with Tukey's multiple comparison test was used to compare the neutralization titers before and after booster vaccination. ***p*-values < 0.001, *p*-value: 0.02. **b** Absolute neutralization titers (left) and fold-reduction (right) for WA-1, BA.1 and BA.1.23 isolates by sera from study participants who experienced breakthrough infection with BA.1. Data for paired sera from 11 participants collected before and after BA.1 infection ($n = 22$) are shown. The error bars represent the geometric mean with 95% confidence intervals. A one-way RM ANOVA with Tukey's multiple comparison test was used to compare the neutralization titers before and after BA.1 breakthrough infection. *p*-values: 0.02, ns not significant. **c** Absolute neutralization titers for each of the isolates (WA-1, BA.1 and BA.1.23) by sera collected from patient P2 before and after BA1.23 infection. The first vertical dotted line (left) represents the time of the third vaccine dose as days relative to the index case. The second vertical dotted line (right) indicates the time of infection with the BA.1.23 variant. **d** Comparison of neutralization fold-change in inhibitory dilution 50% (ID50) measured before and after booster vaccination (left panel, based on 4a) as well as before and after BA.1 (middle panel, based on 4b) and BA.1.23 (right panel, patient 2, based on 4c) breakthrough infection for each of the three viruses tested (WA-1, BA.1 and BA.1.23). Each dot represents IC50 or fold-change data for a specific serum specimen tested by serial dilution in a single replicate experimental setting. GMT denotes mean geometric mean of the inhibitory dilution 50% (ID50) values. The horizontal dotted line represents the limit of detection (10). Samples with neutralization titers below the level of detection were assigned the neutralization value of 5 (equaling to half of the limit of detection) in the ID50 plots. Source data for this figure is provided in the Source Data file.

(e.g., BA.4, BA.5) and subvariants (e.g., BA.2.72.2, BA.4.6/BF.7 (R346T), BA.4.7 (R346S), and BA.5.9 (R346I))[41]. These variants were not circulating at the time of BA.1.23 emergence pointing to convergent evolution in the context of suboptimal immune responses.

We further found that changes in minority viral populations can greatly increase the spectrum of mutations during persistent infection beyond what is observed at the consensus level. Viral 'quasispecies' have been described for pandemic coronaviruses such as SARS-CoV-1, Middle Eastern Respiratory Syndrome Coronavirus (MERS-CoV) and SARS-CoV-2[42–48], but their role remains to be fully elucidated[49]. Interestingly, miSNVs predominantly occurred in conserved regions and remained present in multiple sequential specimens. Only a small subset of these mutations became dominant in later specimens. We speculate that these viral populations enable sub-optimal mutations to persist within the host, increasing the likelihood for the emergence of compensatory mutations that offset their fitness defects.

Despite careful monitoring, we did not encounter new BA.1.23 cases in our routine SARS-CoV-2 surveillance or in the GISAID database between June-September 2022, suggesting that the lineage has not spread further beyond the initial outbreak. This may be due to increased caution by immunocompromised patients to avoid contacts that could increase their risk of infection. We also note that patients P1, P3 and P4 were hospitalized for long stretches of time during their persistent infections, further limiting community exposure. Finally, the limited transmission could reflect a reduced fitness of BA.1.23 compared to contemporary lineages. Notable in this respect is that the emergence of the BA.1.23 lineage coincided with the general displacement of BA.1 by BA.2 sublineages in the NYC area, in particular BA.2.12.1, which dominated SARS-CoV-2 cases during the spring and early summer of 2022. A longer observation period with continued background surveillance will be needed to rapidly identify any potential re-emergence of BA.1.23 in patients other than the ones included in this study.

Our findings add to the accumulating knowledge regarding SARS-CoV-2 persistent infections, and document subsequent forward transmissions. Furthermore, we show that persistent infections can drive emergence of viral variants with the potential to spread to other individuals, some of whom may themselves develop prolonged infections, and thereby establish a pathway for continued virus evolution. This warrants genomic monitoring of persistent infections in particular, and further underscores the need to limit the duration of viral infection. Improved early detection of novel SARS-CoV-2 variants and forward transmission tracking, a better understanding of the selection processes driving SARS-CoV-2 evolution and the role of miSNVs, as well as therapies that limit virus persistence and shedding, are essential to reduce the emergence of highly mutated viral variants in the future.

## Methods

### Molecular SARS-CoV-2 diagnostics
SARS-CoV-2 molecular diagnostic testing was performed in the Molecular Microbiology Laboratories of the MSHS Clinical Laboratory by nucleic acid amplification tests (NAAT) that have been validated for nasopharyngeal (NP), anterior nares (AN) swabs, and saliva specimens. All but one positive sample included in this study were tested using the PerkinElmer® New Coronavirus Nucleic Acid Detection Kit which provides qualitative detection of nucleic acid from SARS-CoV-2. It includes two SARS-CoV-2 targets (ORF1ab, N) and one internal positive control (IC; bacteriophage MS2). Cycle threshold (Ct) values are generated for all three targets and provide a quantitative estimate of the viral load. The only other positive sample that was run on another testing platform was the first specimen of the index case (P1). It was tested as a point of care test (POC) using the cobas® Liat® System (cobas® SARS-CoV-2 & Influenza A/B) assay, and the biospecimen was discarded prior to transfer to the MS-PSP.

### The Mount Sinai Pathogen Surveillance Program (MS-PSP)
The Mount Sinai Pathogen Surveillance Program (MS-PSP) has profiled the evolving landscape of SARS-CoV-2 in New York City (NYC) since the beginning of the pandemic[50,51]. We routinely perform complete viral genome sequencing of randomly selected contemporaneous SARS-CoV-2 positive specimens collected from patients seeking care at our health system. Residual respiratory specimens (NP, AN swabs) were collected after completion of the diagnostic process, as part of the Mount Sinai Pathogen Surveillance Program.

### Human serum samples
We used a panel of sera collected as part of the longitudinal observational PARIS (Protection Associated with Rapid Immunity to SARS-CoV-2) cohort[52]. This study follows health care workers longitudinally since April 2020. Samples were selected based on the study participants different levels of immunity (before and after booster vaccination, as well as before and after Omicron BA.1 infection). Sera collected before and after SARS-CoV-2 booster vaccination ($n = 9$ participants, 18 sera), as well as before and after Omicron BA.1 break-through infections ($n = 11$ participants, 22 sera), were selected for testing in this study (see Supplementary Tables 1 and 2 for infection, vaccine and demographics information). Several serum samples from patient P2 before and after BA.1.23 infection, were available for neutralization assays. The 44 sera (40 PARIS, 4 from P2) from our observational cohorts are unique to this study and are not publicly available.

### Ethics statement
The program was reviewed and approved by the Mount Sinai Hospital (MSH) Institutional Review Board (IRB-13-00981). The detailed investigation into persistent SARS-CoV-2 infections in patients receiving care at

the Mount Sinai Health System was separately reviewed and approved by the MSH IRB (IRB-22-00760). The PARIS study was reviewed and approved by the MSH IRB (IRB-20-03374). All participants signed written consent forms prior to sample and data collection and provided permission for sample banking and sharing. In addition, patient P2 participated in another of our observational studies (IRB-16-01215). They provided written consent prior to biospecimen and data collection.

## RNA extraction and SARS-CoV-2 genome sequencing

RNA was extracted using the Chemagic ™ Viral DNA/RNA 300 Kit H96 (PerkinElmer, CMG-1033-S) on the automated Chemagic ™ 360 instrument (PerkinElmer, 2024-0020) from 300 uL of viral transport media per the manufacturer's protocol. cDNA synthesis followed by whole genome amplification with two custom primer panel sets targeting 1.5 and 2 kb regions across the SARS-CoV-2 genome was performed as previously described[50] with the addition of new oligonucleotides to minimize amplicon dropout for Omicron lineages derived from PANGO lineage B.1.1.529 (Supplementary Table 3). Paired-end (2x150bp) Nextera XT (Illumina, cat. FC-131-1096) libraries prepared from amplicons were sequenced on a MiSeq instrument. SARS-CoV-2 genomes were assembled using our custom Virus Reference-based Assembly Pipeline and IDentification (vRAPID)[53] tool with modifications from our previously described covid assembly pipeline[50]. The final average sequencing depth per genome ranged between ~135k to ~415k reads.

## Analysis of minor nucleotide variants

Minority intrahost single nucleotide variants (miSNVs) were annotated when present in forward and reverse-strand reads of a single sample and at a minimum frequency of 0.1 (10%). A second quality control filter was applied by only including positions for which miSNVs were present in more than one sample from the study (from different time points) or for positions that changed at the consensus level at any point of the investigation.

## Phylogenetic analysis

Global background SARS-CoV-2 sequences were downloaded from the GISAID EpiCoV database (as of 2022-11-07). The GISAID database was queried for the novel mutations observed in P1-P4 sequences to identify their closest matches. Sequence hits were confirmed by their Mash distance[54]. For this, a genome sketch was generated with Mash v.2.3 from the sanitized alignment of the global sequences filtered for lineages BA.1.* as produced by Nextstrain[1,55] v11 for SARS-CoV-2 with default parameters (https://github.com/nextstrain/ncov). This allowed pairwise comparisons of our data with high-quality global sequences with available metadata. Maximum likelihood phylogenetic inferences of the MS-PSP SARS-CoV-2 genomes, including the closest sequence matches and all other BA.* lineage sequences from the MS-PSP surveillance program, were done using an Omicron BA.* and a New York State-focused background with proximity subsampling scheme with Nextstrain under the GTR + G model of nucleotide substitution. Branch support was assessed with the ultrafast bootstrap method with 1000 replicates in IQ-TREE multicore version 2.1.2[56,57]. Clade and lineage assignments were done with Nextclade CLI v3.2 (2021-11-04)[58] and with PANGO-v1.8 (pangolin v3.1.17, pangoLEARN v.2022-04-22)[59,60].

## Recombination analysis

We performed recombination analysis for the BA.1.23 lineage at the consensus level with RIPPLES (Recombination Inference using Phylogenetic PLacEmentS[23]), with default parameters but allowing a minimum of 3 descendants. This analysis was done using a global scale phylogeny generated with UShER[61] containing an extensive collection of public sequences from Genebank, COG-UK and the China National Center for Bioinformation (http://hgdownload.soe.ucsc.edu/goldenPath/wuhCor1/UShER_SARS-CoV-2/). Parsimony score improvements above 7 were considered to reflect true recombination events.

To account for variants present below consensus levels, we considered all samples from patient P1 starting from day 46, when minority variants were detected at more than 10 positions, to day 131, when the highest number of mutations were observed. We used a sliding window of 3 consecutive mutations to identify lineages carrying similar combinations in covSPECTRUM[24]. Search criteria included the global diversity (world filter) between the period of 2021-11-01 (emergence of Omicron) and 2022-09-30 (10 days after the last BA.1.23 sample was identified).

## SARS-CoV-2 viral cultures

Replication-competent SARS-CoV-2 was obtained as previously described[33]. Briefly, Vero-E6 cells expressing transmembrane serine protease 2 (TMPRSS2) (BPS Biosciences, catalog (cat.) no. 78081) and Vero-E6 cells expressing both TMPRSS2 and angiotensin converting enzyme-2 (ACE2) (BEI Resources, catalog (cat.) no. NR-54970) were cultured in Dulbecco's modified Eagle medium containing 10% heat-inactivated fetal bovine serum and 1% minimum essential medium (MEM) Amino Acids Solution, supplemented with 100 U/ml penicillin, 100 µg/ml streptomycin, 100 µg/ml normocin, and 3 µg/ml puromycin. Cell lines were authenticated by the supplier and determined to be free of Mycoplasma contamination by a PCR-based assay (Universal Mycoplasma Detection Kit, ATCC, Cat. 30-1012 K). 200 ul of viral transport media from the nasopharyngeal or anterior nares swab specimen was added to Vero-E6-TMPRSS2 or Vero-E6-TMPRSS2.T2A.ACE2 cells in culture media supplemented with 2% heat-inactivated fetal bovine serum, 100 g/ml normocin, and 0.5 µg/ml amphotericin B. Cultures were maintained for a maximum of 10 days. Culture supernatants were collected and clarified by centrifugation (3739 g for 5 min) upon the appearance of cytopathic effects. Viral cultures were successful for the initial specimens obtained from patients P2, P3 and P4, but failed for all patient P1 specimens. Two representative and sequenced-verified isolates of BA.1.23 were deposited to NIH's BEI resources.

## Titration and concentration of viral isolates

Viral stocks were sequence-verified and then titered using the 50% tissue culture infectious dose ($TCID_{50}$) method on Vero-E6-TMPRSS2 cells. Based on sequencing verification, a viral isolate from P3 (PV56567) was selected for further experiments. Since the initial viral titer of the expanded BA.1. 23 PV56567 viral stock was too low ($4 \times 10^2$ $TCID_{50}$/ml) to perform microneutralization assays, we concentrated the viral isolate 4x using Pierce™ Protein Concentrator PES, 100 K MWCO (Thermo Scientific, Catalog number: 88537) columns as described by the manufacturer. Briefly, culture supernatant was collected upon appearance of a cytopathic effect (CPE), clarified by centrifugation (3739 g, 5 min) and then concentrated. Titers after concentration were $2.25 \times 10^5$ $TCID_{50}$/ml.

## Micro-neutralization assays with replication-competent SARS-CoV-2 isolates

Sera collected from three different groups of study participants were used to assess the neutralization of wild type (WA1), BA.1, and BA.1.23 SARS-CoV-2 isolates. The first panel of samples includes sera collected before and after mRNA SARS-CoV-2 booster vaccination ($n = 9$, PARIS cohort), the second panel includes sera collected before and after BA.1 Omicron break-through infection ($n = 11$, PARIS cohort). The last series includes longitudinal samples from P2, the first of the forward transmission cases, collected prior as well as after the infection with BA.1.23.

Sera from study participants were serially diluted (three-fold) from a starting dilution of 1:10 using infection media consisting of minimum essential media (MEM, Gibco) supplemented with 2 mM L-glutamine, 0.1% sodium bicarbonate (w/v, Gibco), 10 mM 4-(2-hydroxyethyl)-1-piperazineethanesulfonic acid (HEPES, Gibco), 100 U/ml penicillin, 100 µg/ml streptomycin (Gibco) and 0.2% bovine serum albumin (MP Biomedicals). Diluted sera were incubated for one hour

with 10,000 $TCID_{50}$ of the three different viruses at room temperature. After the incubation, 120 μl of the serum-virus mix were transferred to Vero-E6-TMPRSS2 (plated in 96 well plates the prior day) and incubated at 37 °C, 5% $CO_2$ for one hour. The serum-virus mix was removed and 100 μl/well of the diluted sera were added in addition to 100 μl/well of MEM 2% FBS. The plates were incubated for 48 h at 37 °C incubation. The cells were fixed (200 μl/well, 10% formaldehyde solution) overnight at 4 °C prior to staining with biotinylated monoclonal SARS nucleoprotein antibody 1C7C7 (Millipore Sigma, cat. no. ZMS1075) at a concentration of 1 μg ml−1, followed by secondary staining with HRP-conjugated streptavidin (Thermo Fisher Scientific) at a 1:2000 dilution as previously described[33]. All procedures above were performed twice in the Biosafety Level 3 (BSL-3) facility at the Icahn School of Medicine at Mount Sinai following approved standard safety guidelines. The nucleoprotein staining was performed as previously described[33]. All commercial antibodies were validated by their manufacturers and were titrated in the lab to determine optimal concentration for experimentation. In-house biotinylated 1C7C7 monoclonal antibody was validated in cells infected with WT SARS-CoV-2, BA.1 and BA.1.23 viral isolates.

### Statistics

Statistical analyses were performed using Prism 9 software (GraphPad). A one-way ANOVA with Tukey's multiple comparisons test was used to compare the neutralization titers for the three viruses.

### Reporting summary

Further information on research design is available in the Nature Portfolio Reporting Summary linked to this article.

## Data availability

Complete genome sequences for the viral isolates cultured from nasal swabs (BA.1 and BA.1.23) are available in GenBank (accession numbers ON220548, ON220539, ON196014, ON193425, ON220529, ON220571, ON220533, ON934538, ON934577, ON934030, ON934607, ON854465, ON619382). RNA-seq data are available in the Sequence Read Archive (SRA), submission SUB12865927 under BioProject number PRJNA623586 (BioSample accession numbers SAMN33273632 - SAMN33273655). Datasets generated and/or analysed during the current study are appended as supplementary data. Source data are provided with this paper.

## Code availability

The vRAPID pipeline was developed for reference-based viral genome assembly of short-read sequencing data. The custom code is at vRAPID [https://zenodo.org/record/7829342].

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

## Acknowledgements

We thank the Mount Sinai Hospital and School Leadership, in particular Dr. David Reich, Dr. Dennis Charney, and Dr. Carlos Cordon-Cardo, for their ongoing support of the MS-PSP throughout the pandemic. We thank the laboratory technologists and staff in the Molecular Microbiology Laboratories and the Rapid Response Laboratories of the Mount Sinai Health System since without their assistance and help, none of this surveillance work would be possible. We thank Dr. R. A. Albrecht for oversight of the conventional BSL-3 biocontainment facility at Mount Sinai, which makes our work with replication-competent SARS-CoV-2 isolates possible. We gratefully acknowledge the authors and originating and submitting laboratories of sequences from GISAID's EpiCoV (www.gisaid.org) that were used as background for our phylogenetic inferences. The work reported was, in part, supported by a contract from the National Institute of Allergy and Infectious Diseases (75N93021C00014, Option 12 A, to V.S. and H.v.B.) awarded to the Center for Research on Influenza Pathogenesis and Transmission and by an Option to the Collaborative Influenza Vaccine Innovation Centers (CIVIC) contract 75N93019C00051 (to F.K. and V.S.) as part of the SARS-CoV-2 Assessment of Viral Evolution (SAVE) Program, a contract from the National Institute of Allergy and Infectious Diseases (HHSN272201400008C, Option 20, to V.S., F.K., and H.v.B.) awarded to the Center for Research on Influenza Pathogenesis, and awards (S10OD026880 and S10OD030463, to ISMMS) from the NIH Office of Research Infrastructure Programs. Additional support was provided by R21AI169280 (to V.S.) and U19AI168631 (to V.S., F.K., and H.v.B.). The Mount Sinai Pathogen Surveillance Program is supported in part by Institutional funds from the Icahn School of Medicine as well as the Mount Sinai Hospital (to E.M.S., V.S. and H.v.B.). This work was also supported in part through the computational and data resources and staff expertise provided by Scientific Computing and Data at the Icahn School of Medicine at Mount Sinai and supported by the Clinical and Translational Science Awards (CTSA) grant UL1TR004419 from the National Center for Advancing Translational Sciences (to the Icahn School of Medicine at Mount Sinai).

## Author contributions

Conceptualization: A.S.G.-R., H.A., E.M.S., V.S., H.v.b. Sample acquisition: S.S., G.P., J.P., A.A., A.R., C.C., D.F., L.A.S., C.G., K.S., J.D.R., R.B., P.S., A.P.-M., PSP-PARIS Study Group. Sequencing and

genome assembly: A.vd.G., K.F., Z.K., J.Z., B.A., R.S. Methodology: A.S.G.-R., H.A., N.A., J.M.C., F.K. Investigation: A.S.G.-R., H.A., J.M.C., E.M.S., V.S., H.v.B. Visualization: A.S.G.-R., H.A., V.S., H.v.B. Funding Acquisition: E.M.S., V.S., H.v.B. Project administration: K.S., E.M.S., V.S., H.v.B. Supervision: E.M.S., V.S., H.v.B. Writing – First draft: A.S.G.-R., H.A., H.v.B. Writing – Review and editing: A.S.G.-R., H.A., J.M.C., F.K. E.M.S., V.S., H.v.B.

## Competing interests

The Icahn School of Medicine at Mount Sinai has filed patent applications relating to SARS-CoV-2 serological assays (U.S. Provisional Application Numbers: 62/994,252, 63/018,457, 63/020,503 and 63/024,436) and NDV-based SARS-CoV-2 vaccines (U.S. Provisional Application Number: 63/251,020) which list Florian Krammer as co-inventor. Viviana Simon is also listed on the serological assay patent application as co-inventor. Patent applications were submitted by the Icahn School of Medicine at Mount Sinai. Mount Sinai has spun out a company, Kantaro, to market serological tests for SARS-CoV-2. Florian Krammer has consulted for Merck and Pfizer (before 2020), and is currently consulting for Pfizer, Third Rock Ventures, Seqirus and Avimex. The Krammer laboratory is also collaborating with Pfizer on animal models of SARS-CoV-2. Robert Sebra is currently a paid consultant and stockholder of GeneDx.

## Additional information

[1]Department of Genetics and Genomic Sciences, Icahn School of Medicine at Mount Sinai, New York, NY 10029, USA. [2]Department of Microbiology, Icahn School of Medicine at Mount Sinai, New York, NY 10029, USA. [3]Center for Vaccine Research and Pandemic Preparedness (C-VaRPP), Icahn School of Medicine at Mount Sinai, New York, NY 10029, USA. [4]Division of Infectious Diseases, Department of Medicine, Icahn School of Medicine at Mount Sinai, New York, NY 10029, USA. [5]Graduate School of Biomedical Sciences, Icahn School of Medicine at Mount Sinai, New York, NY 10029, USA. [6]Icahn Genomics Institute, Icahn School of Medicine at Mount Sinai, New York, NY 10029, USA. [7]Black Family Stem Cell Institute, Icahn School of Medicine at Mount Sinai, New York, NY 10029, USA. [8]The Global Health and Emerging Pathogens Institute, Icahn School of Medicine at Mount Sinai, New York, NY 10029, USA. [9]Department of Pathology, Molecular, and Cell-Based Medicine, Icahn School of Medicine at Mount Sinai, New York, NY 10029, USA. [10]Centro de Investigaciones en Microbiología y Biotecnología-UR (CIMBIUR), Facultad de Ciencias Naturales, Universidad del Rosario, Bogotá, Colombia. [11]The Global Health Emerging Pathogens Institute, Icahn School of Medicine at Mount Sinai, New York, NY 10029, USA. ✉e-mail: Emilia.Sordillo@mountsinai.org; viviana.simon@mssm.edu; harm.vanbakel@mssm.edu

## PARIS/PSP study group

**Giulio Kleiner[2], Dalles Andre[2], Katherine F. Beach[2], Maria C. Bermúdez-González[2], Gianna Cai[2], Neko Lyttle[2], Lubbertus C. F. Mulder[2], Annika Oostenink[2], Ashley Beathrese T. Salimbangon[2], Gagandeep Singh[2], Morgan van Kesteren[2], Brian Monahan[2], Jacob Mauldin[2] & Mahmoud Awawda[12]**

[12]Environmental Medicine and Public Health, Icahn School of Medicine at Mount Sinai, New York, NY 10029, USA.

