## [Peer Review File · Nature Communications]

Sequential intrahost evolution and onward transmission of SARS-CoV-2 variantsReviewers' Comments:

Reviewer #1:

Remarks to the Author:

In this manuscript, Gonzalo-Reiche et al. document the case of an immunosuppressed, chronically infected SARS-CoV-2 patient (P1) over the course of 81 days along with indirect evidence of forward transmission from this individual to 3 other patients (P2, P3, and P4) and into the broader community (2 GISAID sequences from NYC). In terms of detailing viral evolutionary dynamics in chronically infected individuals, this work adds to many existing SAR-CoV-2 studies, without profoundly new insights. The real novelty of this work is that it provides sequence-derived evidence of forward transmission of a highly evolutionarily divergent viral lineage from this chronically infected individual (P1) to other individuals. This is important to demonstrate because VOCs such as Omicron and Alpha have been hypothesized to have evolved in chronically infected individuals but there has been a lack of case studies that have demonstrated that viral genotypes that have evolved in these individuals can lead to successful forward transmission.

While the work presented in this manuscript is important and the analyses presented appear to be largely thoroughly done, I think the work would benefit from a major reorganization. Currently, the results section begins with a detailed description of the consensus level spike gene viral substitutions that occurred in patient P1. The next section then uses those substitutions as evidence for forward transmission to patients P2-P4 and the broader community. Only then do the authors point to a phylogeny (Fig 2A) briefly, prior to detailing the evolution of the viral populations in patients P1-P4 below the consensus level.

It seems to me that reorganizing the manuscript to first present figures to conclusively demonstrate forward transmission based on whole-genome consensus sequences would be the results to start with. This would include the current Figure 2A (which would become Figure 1A). A second, time-aligned phylogeny here would also be very informative as a Figure 1B, as it would help reconstruct transmission times (based on tMRCAs/internal node times). Separate tables for P1-P4 and the 2 GISAID sequences could then list consensus-level nucleotide substitutions (ideally relative to P1 (d23), for clarity, since we don't care about the substitutions that have occurred since the Wuhan reference strain). These tables could indicate which of these substitutions were synonymous and which nonsynonymous (along with amino acid changes). Then present the below-the-consensus findings and bring those into the fold of whether those tell you anything more about the timing of transmission, or potentially bottlenecks between individuals. And then analyze the evolutionary dynamics in terms of selection and phenotypic changes that occurred (this would include both substitutions and 'mSNVs'). I'm suggesting this reorganization such that the questions of interest are sequentially addressed:

- Did onward transmission occur from chronically infected patient P1?
- If yes, when did onward transmissions occur?
- How did the virus evolve within P1 (selection, immune escape, etc.)?
- Which phenotypes did the transmitted virus carry?

I think most of the analyses are there in the manuscript already; there is just considerable effort needed on the part of the reviewer to piece the parts together.

Other major comments:

Some of the samples were NP samples and some were AN samples, but, with the exception of Figure 3, none of the other analyses indicate this. Could there be compartmentalization occurring between these sampling sites, such that effectively analyzing the sequence data as though they came from the same site may not be appropriate?

Line 116-117: more metadata here if possible would be really helpful and strengthen the results considerably. Is there documentation that these three secondary infections (patients) had direct contact with P1? Or potentially indirect contact through use of the same room?

According to Figure 2A, P3's day 131 consensus sequence appears to be highly divergent – even relative to that same patient's day 117 and day 112 sequences. This doesn't seem right. I know the authors argue that this is not recombination, but more analyses to make sure that the results here are accurate, and if they appear to be, how to interpret these dramatic changes would be good. I think to do this properly, one would need to infer haplotypes in the viral population based on the read data,

and then look at changes in haplotype frequencies over time.

One critical comment is about the availability of the sequence data. According to the Data Availability Statement, only nasal swab consensus samples have been made available (on GISAID). I assume these include NP and AN samples? For reproducibility, the deep sequencing data also need to be made available, and the accession numbers of the sequences and the SRA project numbers need to also be provided.

Minor comments:

Lines 93-106: it's unclear to me which of these identified mutations were at the consensus level versus fixed vs present at low frequencies. I think, given the reference to Figure 1, that this analysis is all consensus level.

Line 103: The conclusion that 'two different viral populations emerged' is confusing to me if all the analyses are at the consensus level – having those three different substitutions at the consensus level doesn't necessarily indicate to me that two different viral populations emerged. To conclude this, it seems that below-the-consensus analyses would have to be done (with haplotype reconstruction).

Lines 104-106 indicate that there were nonsynonymous (+1 synonymous) mutations outside of spike, but those results I don't see in Figure 1.

I am trying to wrap my head around Figure 1... These are SNVs relative to which reference? If a position goes from blue line to red line (e.g. E96D), what does that mean? Are all unlabeled SNVs synonymous and labeled nonsynonymous (I don't think this is the case, but if that's not the case, is there information in this figure that denotes which type the SNV is?) What does a bold line signify vs an unbolded line? Day 48, FCS region: is there an error in plotting, i.e. should the red line be under the red line for day 40? Based on Figure 2A, it is an error in plotting.

Lines 88-91: 'Some of these escape mutations in BA.1.23 are now signature substitutions in emerging Omicron lineages such as BA.2.75.2, indicating that persistent viral replication in the context of suboptimal immune responses is an important driver of SARS-CoV-2 diversification' This statement I think is an overreach. How do the observed convergent mutations between BA.1.23 and BA.2.75.2 indicate that 'persistent viral replication in the context... is an important driver of SARS-CoV-2 diversification'?

Line 112: E484V (rather than A484V?) – at least according to Figure 1.

Line 114: 'Although viral isolation failed for specimens available from the index case..' I don't understand. Wasn't the virus isolated from P1?

Lines 116-117: Here, it seems that the forward transmissions were inferred based only on the common nonsynonymous substitutions found in spike. Why present this work this way, rather than by presenting Figure 2A first (whole-genome analysis, and using both nonsynonymous and synonymous variation)? Please see my main comment about reorganization.

Line 120: could you clarify how differing in age and gender from the 4 cases (P1-P4) provides an indication that there was independent but limited community spread of this subvariant?

Mapping both synonymous and nonsynonymous substitutions on to the phylogeny (current Figure 2A; or a time-aligned version of this phylogeny) would be helpful

Figure 2B: Are mSNV frequencies also potentially available for the GISAID sequences? (I.e., are there short read data available in the SRA that correspond to these GISAID consensus sequences?)

Figure 2B: Rather than red and blue for in spike vs outside of spike, it would be more helpful to color by nonsynonymous vs synonymous (vertical lines could denote spike region)

Instead of introducing the term 'mSNV', why not called it an 'iSNV' (e.g., McCrone et al. eLife)?

Line 185: how do we see this in Figure 2A?

In sum, this is an important case study that documents limited forward transmission of a highly divergent SARS-CoV-2 lineage that evolved in a chronically infected individual. The impact and interpretability of this work could be considerably improved from a restructuring of the manuscript. Beyond this, there are several other major comments (above) that, if addressed, would strengthen the manuscript.

Reviewer #2:

Remarks to the Author:

Severely immunocompromised patients are at risk for severe and prolonged SARS-CoV-2 infection, and as a result, are an important potential source of viral mutation and development of variants. The authors characterize genetic mutations that occur over time, and attempt to demonstrate forward spread, which would be an important contribution to the field. However, there lacks data that clearly supports transmission of novel strains. Additionally, clinical details provided could be further clarified and link between antivirals and emergency of mutations should not be overstated.

Results, page 5: Authors characterize amino acid substitutions that developed in SARS-CoV-2 strain of patient 1. What is the baseline rate of amino acid substitutions of SARS-CoV-2 to help determine if persistent infection in an immunocompromised host is driver of evolution?

Results, page 5: The authors reference isolating variants with shared amino acid substitutions, all of whom were hematologic malignancy patients. Was actual cluster analysis (time and location) performed to help confirm hospital transmission? If not, do not have enough data to state this is an outbreak.

Results, page 7-8: Patient received non-EUA-approved courses of therapies: e.g. 3-4 weeks of Paxlovid. What was the route of obtaining these therapies?

Results, page 8: Authors make multiple states about therapies received and subsequent detection of mutations or lack of persistent infection that may not be causative.

Discussion, page 11: Provide reference for low antibody levels being risk factor for immune escape mutations

Antivirals were not studies looking at outcome of "eliminating persistent infection" but rather for reducing severity of infection. In Figure 3, it appears that cycle threshold of patients did increase (lower viral load) after receipt of antivirals?

Authors use the term "fully vaccinated" but do not provide details on which vaccines (we know that mRNA vaccines are more immunogenic than adenovirus vector, for example) and how many doses.

Reviewer #3:

Remarks to the Author:

Gonzalez-Reiche et al analysed SARS-CoV-2 persistent infections in immunocompromised patients. They described the emergence, transmission and subsequent evolution of the new Omicron sub lineage BA.1.23 in patients with persistent SARS-CoV-2 infection and replication. They observed that the initial substitutions were within the spike but continued replication led to substitutions in other viral proteins. The authors also showed that BA.1.23 variant was more resistant to neutralising antibodies induced post-booster vaccination or after BA.1 breakthrough infection compared to BA.1 and the ancestral Wuhan strain.

Understanding how SARS-CoV-2 variants emerge and further evolve in immunosuppressed hosts is crucial to develop strategies to treat these individuals in order to prevent the emergence of variants. This paper is highly relevant. The methodology looks appropriate to me. The paper is well written and the figures are very clear. I only have a few comments. Maybe a few additional points below-mentioned could be added to the discussion or clarified.

Major Comments:

1) Line 113: when the authors speak about transmission, do they mean that these patients had some contact(s) at the hospital? Are there any evidences they were in the same unit in the same time?

- 2) Line 117: do the authors mean B-cell and T-cell deficiencies in all patients ? Or only B-cell deficiencies ?
- 3) Lines 181-182: did the authors try to discriminate vaccine-induced and infection-induced antibodies by measuring anti-N ? Even though I presume IVIg may also bind to N. More globally, I was wondering whether the authors compared both anti-S and anti-N in comparison to see if residual antibodies of different specificities were measured.
- In addition, I don't see Figure 2C mentioned in the text. It is probably 3B.
- 4) Line 198 and line 205: the data suggest that a suboptimal level of antibodies may lead to a selective pressure. Do the authors mean these antibodies were not functional (neutralising, Fc receptor function) ? I was wondering whether it was only related to the magnitude or also a lack of functionality.
- In addition, related to the use of MAb treatment, is there a way to play with the dose and/or length of treatment to make sure the benefit outweighs the risk of selective pressure ?
- 5) Did the authors analyse if currently available MAb-based therapies could neutralise BA.1.23 ?
- 6) Line 262: Is there any role of T cells in the intrahost evolution ? Can selection be also driven by T cell escape ? Or is it only related to antibody response ? Do the authors have any T cell data in these patients ?
- 7) Are there any forms of immunosuppression which are more susceptible to lead to the emergence of variants and mutations leading to escape patterns ?

Minor comments:

Figure 3: legends B and C were inverted

Line 383: typo "different time points"

Extended data Figure 1: what does n mean ?

Extended data Figures 2 and 3: could the authors use the same colors for New and BA.1.23 in each figure (Extended Figures 2, 3 and 4) ?

Reviewer #4:

Remarks to the Author:

Major comments

The epidemiological contexts linking the index case to the other individuals should be stated. In the relevant Results sub-section (lines 108- 128), the authors describe forward transmission from the index case (P1) to five other individuals, comprising three immunocompromised patients (P2 P3, P4) and two community members (GISAID S1 and S2), based on the shared presence of a unique combination of 7 spike protein mutations.

Is there available information on how the other individuals came into contact with the index case, such as whether the three patients shared the same ward?

Did the two additional individuals from the NYC area come into contact with the index case or is there a possibility they were contaminated elsewhere?

This would help the reader understand the transmissibility of the BA.1.23 variant.

Line 214-215 Why were these 7 mutations selected to be the defining mutations of BA.1.23 out of all detected mutations, not including minority variants (Figure 1)?

Line 203-204. P4 had low levels of antibody titres at the time of the first positive PCR although they were unvaccinated. Could this be an indication of a previous Covid infection? Are past infection histories of the other cases known?

Minor comments

Line 45-46. I find this sentence to be misleading as it sets up the reader to imagine forward

transmissions from three index cases. While it is true the study includes three individuals with persistent infections, the fact that there was only one source of forward transmission affects how the transmissibility of the emerging variant BA.1.23 is perceived.

Figure 1. The legend should explain why some mutations are in bold.

Figure 3C and 3B legends should be swapped.

Line 160 "Figure. B2".

Line 383 "time pointes".

Figure 4 Neutralisation titre panels. Presently, the GMT values are too easily confused as X-axis values. It might clarify the graph to remove the X-axis ticks and place the GMT values in a table to dissociate them from the X-axis.

Extended Data Figure 2: Do the authors mean diamonds instead of triangles?

Point by point responses to each of the reviewers

We thank the reviewers for their thoughtful and constructive evaluations, and we have revised the manuscript to address their comments.

Of note, we added data from three additional nasopharyngeal swab specimens from patient P3 that were collected after the submission of the original manuscript. No additional BA.1.23 cases were detected in our health system or globally after the latest sequenced specimen was captured from P3, who passed 271 days after the initial identification of BA.1.23. We believe that our precision surveillance approach has helped to limit the spread of the BA.1.23 lineage.

Reviewer #1:

In this manuscript, Gonzalo-Reiche et al. document the case of an immunosuppressed, chronically infected SARS-CoV-2 patient (P1) over the course of 81 days along with indirect evidence of forward transmission from this individual to 3 other patients (P2, P3, and P4) and into the broader community (2 GISAID sequences from NYC). In terms of detailing viral evolutionary dynamics in chronically infected individuals, this work adds to many existing SAR-CoV-2 studies, without profoundly new insights. The real novelty of this work is that it provides sequence-derived evidence of forward transmission of a highly evolutionarily divergent viral lineage from this chronically infected individual (P1) to other individuals. This is important to demonstrate because VOCs such as Omicron and Alpha have been hypothesized to have evolved in chronically infected individuals but there has been a lack of case studies that have demonstrated that viral genotypes that have evolved in these individuals can lead to successful forward transmission.

R1.1. While the work presented in this manuscript is important and the analyses presented appear to be largely thoroughly done, I think the work would benefit from a major reorganization. Currently, the results section begins with a detailed description of the consensus level spike gene viral substitutions that occurred in patient P1. The next section then uses those substitutions as evidence for forward transmission to patients P2-P4 and the broader community. Only then do the authors point to a phylogeny (Fig 2A) briefly, prior to detailing the evolution of the viral populations in patients P1-P4 below the consensus level.

It seems to me that reorganizing the manuscript to first present figures to conclusively demonstrate forward transmission based on whole-genome consensus sequences would be the results to start with. This would include the current Figure 2A (which would become Figure 1A). A second, time-aligned phylogeny here would also be very informative as a Figure 1B, as it would help reconstruct transmission times (based on tMRCAs/internal node times). Separate tables for P1-P4 and the 2 GISAID sequences could then list consensus-level nucleotide substitutions (ideally relative to P1 (d23), for clarity, since we don't care about the substitutions that have occurred since the Wuhan reference strain). These tables could indicate which of these substitutions were synonymous and which nonsynonymous (along with amino acid changes). Then present the below-the-consensus findings and bring those into the fold of whether those tell you anything more about the timing of transmission, or potentially bottlenecks between individuals. And then analyze the evolutionary dynamics in terms of selection and phenotypic changes that occurred (this would include both substitutions and 'mSNVs').

I'm suggesting this reorganization such that the questions of interest are sequentially addressed:

- Did onward transmission occur from chronically infected patient P1?
- If yes, when did onward transmissions occur?
- How did the virus evolve within P1 (selection, immune escape, etc.)?
- Which phenotypes did the transmitted virus carry?

I think most of the analyses are there in the manuscript already; there is just considerable effort needed on the part of the reviewer to piece the parts together.

Answer: We appreciate the suggestion by this reviewer, but we feel that starting the results with a presentation of all specimens and cases would not reflect the chronology of events, which included an initial stage of emergence of the BA.1.23 lineage, followed by forward transmission and further evolution of the lineage in recipients that developed additional persistent infections. We also note that none of the other reviewers have commented on the order in which the results are presented.

To address questions regarding the timing of onward transmissions, we have constructed a time-aligned phylogeny to help reconstruct transmission times, now shown as an insert on Figure 2A and displaying the estimated TMRCAs with 95% confidence intervals. We also now discuss the potential for direct or indirect contact between patients P1, P2, P3 and P4 while receiving care in our health system, based on admission/discharge/transfer records.

Other major comments:

R1.2: Some of the samples were NP samples and some were AN samples, but, with the exception of Figure 3, none of the other analyses indicate this. Could there be compartmentalization occurring between these sampling sites, such that effectively analyzing the sequence data as though they came from the same site may not be appropriate?

Answer: We have added sample type annotations to Figure 2 to ensure that the same information is consistently represented throughout the manuscript. We found no strong evidence for compartmentalization as there is no grouping of genotypes by sample origin. However, after the addition of three new specimens for patient P3 we did note that there was a notable decrease in minor intrahost SNVs (miSNVs) and increase in consensus mutations in the AN specimen collected from P3 on day 131, compared to the earlier and later NP specimens. We have now included this information on lines 173-175.

R1.3: Line 116-117: more metadata here, if possible, would be really helpful and strengthen the results considerably. Is there documentation that these three secondary infections (patients) had direct contact with P1? Or potentially indirect contact through use of the same room?

Answer: Based on the pattern of mutations in the viral genome we suspect the initial transmission(s) from P1 occurred in an 18-day period between day 64 and day 82. During this time, P2 (10 days overlap) and P4 (18 days overlap) were admitted to same unit (unit A) as P1. P2 and P4 stayed in neighboring rooms, which were separated by 6 rooms from P1. On day 80, P4 was transferred to another room in a different unit (unit B), which was separated by 1 room from P3. Patients P3 and P4 overlapped in unit B for two

days until P4 was discharged on day 82. As such, there was a potential for direct or indirect contact between all four patients.

A potential sequence of events based on the available data is that BA.1.23 initially transmitted from P1 to P2 and/or P4 in Unit A, with possible secondary transmissions between P2 and P4. Onward transmission from P4 to P3 then took place in unit B. In this scenario, both P3 and P4 would have been infected with BA.1.23 for three weeks before their initial positive tests. While this is well beyond documented incubation times of SARS-CoV-2, we note that both patients developed persistent infections lasting between 1-4 months. As such, it is possible that the onset of persistent infection in these patients occurred earlier, during the window of potential contact. An alternative scenario is that P1 transmitted to P2 on unit A between day 64 and day 72, before P2 was discharged on day 72. P3 and P4 then had later exposures before both patients were re-admitted to the hospital.

As P3 and P4 were not tested for SARS-CoV-2 during the three-week window between potential contact and their initial positive tests, we cannot determine which scenario is the most likely. We therefore chose to provide a more limited assessment of potential patient contacts based on the unit admission history in the revised manuscript (Lines 126-138).

R1.4: According to Figure 2A, P3's day 131 consensus sequence appears to be highly divergent – even relative to that same patient's day 117 and day 112 sequences. This doesn't seem right. I know the authors argue that this is not recombination, but more analyses to make sure that the results here are accurate, and if they appear to be, how to interpret these dramatic changes would be good. I think to do this properly, one would need to infer haplotypes in the viral population based on the read data, and then look at changes in haplotype frequencies over time.

Answer: We were also intrigued by the observation of the highly divergent sequence on day 131. To confirm this was not a technical artifact, the specimen was sequenced twice. In addition, we successfully isolated the virus in tissue culture and sequenced the viral isolate. The replicate sequences from the original isolate and virus isolates (both passage 0 and passage 1) contained the same mutations at the consensus level. The viral isolates also contained a mutation observed at low frequency in the original specimen at nucleotide position 4391 (C-A). These data indicate strongly that the constellation of mutations observed in the original specimen are real and not artefacts introduced during PCR amplification or the sequencing process. (See Nextclade results below).

ID	Sequence name	QC	Clade	Pango lineage (Nextclade)	Unaligned	Mut.	non-ACGTN	No.	Cov.	Caps	Hts.	FS	SC	Nucleotide sequence
0	P3_d131_original	100	21K (Omicron)	BA.1.23	BA.1.23	70	0	0	99.3%	39	9	0	0	
1	P3_d131_original (2)	100	21K (Omicron)	BA.1.23	BA.1.23	70	0	0	99.3%	39	9	0	0	
2	P3_d131_Passage_1	100	21K (Omicron)	BA.1.23	BA.1.23	71	0	0	99.3%	39	9	0	0	
3	P3_d131_Passage_2	100	21K (Omicron)	BA.1.23	BA.1.23	72	0	0	99.6%	39	9	0	0	

The lack of spanning read information to link distal mutations complicates reconstruction of haplotypes from short read sequencing data. However, we performed additional analyses to rule out recombination. First, we performed recombination analysis at the consensus level with RIPPLES [PMID: 35952714], using default parameters except that we reduced the minimum number of descendants from 10 to 3 to account for our sample size (n=24 with a maximum of 13 genomes per patient). This analysis was done using a global scale phylogeny generated with USHER from a comprehensive collection of unique public sequences from Genbank, COG-UK and the China National Center for Bioinformatics

(http://hgdownload.soe.ucsc.edu/goldenPath/wuhCor1/USHER_SARS-CoV-2/). The RIPPLES analysis did not identify recombination events for P3 genomes with parsimony score improvements above 7, which are more likely to reflect true recombination events.

We also queried the combination of nucleotide substitutions observed in the sequences for P3 containing minority variants from day 46 to 131. For this we used a sliding window of 3 consecutive mutations to identify lineages carrying similar combinations in covSPECTRUM (<https://cov-spectrum.org/>). Search criteria included the global diversity between the period of November 2021 (emergence of Omicron) and September 2023 (last known detections of BA.1.23). From this analysis, only a handful of queries returned matches with less than 10 sequences identified worldwide. We do not believe these consecutive genotype matches to be valid because: 1) they were very rare; 2) inconsistent with recombination breakpoints, 3) found only outside the US, and/or 4) they pre-dated the appearance of the minority variants in P3 by >3 months.

Patient day	Query	Overall Proportion	Number of Sequences	Country	Lineage	Date of first sequence	Date of last sequence	Date of last sequence in the US
P3 d108	C13458T, G15451A, C19164T	0.00%	7	Thailand, Poland, US, Malaysia	B.1.617.2	2021-11-08	2022-01-03	2021-12-13
P3 d108	G15451A, C19164T, C23604G	0.05%	4248	Malaysia, US, UK, Germany, France	B.1.617.2	2021-11-01	2022-05-09	2022-02-14
P3 d108	C19164T, C23604G, G24368T	0.00%	8	Austria, Poland, Costa Rica, Croatia, Germany	B.1.617.2	2021-11-01	2022-01-03	N/A
P3 d112	C22713T, C23604G, G24368T	0.00%	1	Peru	AY.102	2021-11-08	2021-11-08	N/A
P3 d112	T8278C, C13458T, G15451A	0.00%	3	US	AY.3	2022-11-29	2022-12-13	2022-12-13
P3 d112	C13458T, G15451A, C15924T	0.00%	4	Germany, US, Switzerland	AY.121.1, AY.3	2022-11-22	2021-12-20	2022-11-22
P3 d131	C13458T, G15451A, C22323T	0.00%	1	Belgium	AY.4	2021-12-06	2021-12-06	N/A

P3 d131	T27384C, C27509T, G28541T	0.00%	8	Sweden, UK	BA.2	2022-05-02	2022-07-25	N/A
---------	---------------------------------	-------	---	---------------	------	------------	------------	-----

In summary, we did not identify contemporary lineages with mutations seen in the P3 d131 consensus sequence that could be clearly identified as a source of recombination with the BA.1.23 lineage. Considering the progressive accumulation of intrahost minority variants over time, we consider recombination an unlikely scenario. We have added the details of this analysis to the revised manuscript in the results (lines 181 to 188).

R1.5. One critical comment is about the availability of the sequence data. According to the Data Availability Statement, only nasal swab consensus samples have been made available (on GISAID). I assume these include NP and AN samples? For reproducibility, the deep sequencing data also need to be made available, and the accession numbers of the sequences and the SRA project numbers need to also be provided.

Answer: All RNA-seq data and relevant metadata have now been deposited to the NIH Sequence Read Archive (SRA) submission SUB12865927, under accession number SAMN33273632- SAMN33273655. The “Data Availability” section has been updated accordingly.

Minor comments:

R1.6: Lines 93-106: it’s unclear to me which of these identified mutations were at the consensus level versus fixed vs present at low frequencies. I think, given the reference to Figure 1, that this analysis is all consensus level.

Answer: This is correct. All mutations discussed in the above-mentioned paragraph were observed at the consensus level. We have clarified this in the text: *“Over a 12-week period, we documented the accumulation of nine amino acid substitutions in the spike protein ... (Fig. 1) within the same patient at the consensus level.”* (lines 96-99). We also made the distinction to the number of changes outside spike clearer.

R1.7: Line 103: The conclusion that ‘two different viral populations emerged’ is confusing to me if all the analyses are at the consensus level – having those three different substitutions at the consensus level doesn’t necessarily indicate to me that two different viral populations emerged. To conclude this, it seems that below-the-consensus analyses would have to be done (with haplotype reconstruction).

Answer: The reviewer is correct. From Figure 1 alone it cannot be concluded that two different viral populations emerged. This remark was to emphasize that additional mutations were observed though the remainder of the infection. We have reworded this sentence as *“During the following weeks additional mutations emerged; samples from this period contained shared (L455W) as well as distinct signature mutations (E96D on day 72, S477D on day 81).”* Line 102-104.

R1.8: Lines 104-106 indicate that there were nonsynonymous (+1 synonymous) mutations outside of spike, but those results I don’t see in Figure 1.

Answer: We have now added a reference to Extended Data Figure 1 that contains the whole genome alignment where the synonymous mutations are included.

R1.9: I am trying to wrap my head around Figure 1... These are SNVs relative to which reference? If a position goes from blue line to red line (e.g. E96D), what does that mean? Are all unlabeled SNVs synonymous and labeled nonsynonymous (I don't think this is the case, but if that's not the case, is there information in this figure that denotes which type the SNV is?) What does a bold line signify vs an unbolded line? Day 48, FCS region: is there an error in plotting, i.e. should the red line be under the red line for day 40? Based on Figure 2A, it is an error in plotting.

Answer: All comparisons are relative to the ancestral Wu-1 reference strain, with BA.1 signature mutations shown in blue and the novel mutations that emerged in patient P1 shown in red. The colorings of the alignment are indicated in the legend and in the figure itself. In the original figure, the red lines representing novel mutations were thicker than the blue lines representing the BA.1 SNVs. To avoid confusion, we have now made the thickness of the line consistent. Please note, however, that the close separation between some consecutive mutations may give the appearance of a single thicker line. In these instances, we further separated the amino acid changes at the bottom of the alignment.

R1.10: Lines 88-91: 'Some of these escape mutations in BA.1.23 are now signature substitutions in emerging Omicron lineages such as BA.2.75.2, indicating that persistent viral replication in the context of suboptimal immune responses is an important driver of SARS-CoV-2 diversification' This statement I think is an overreach. How do the observed convergent mutations between BA.1.23 and BA.2.75.2 indicate that 'persistent viral replication in the context... is an important driver of SARS-CoV-2 diversification'?

Answer: We reworded this statement in the revised manuscript. We now note that BA.1.23 mutations have been found in later lineages, and state that our study adds further evidence for the role of persistent infections in the diversification of SARS-CoV-2, as follows:

"Some of the escape mutations in BA.1.23 have also emerged in later Omicron lineages such as BA.2.75.2. Overall, our results indicate that persistent viral replication in the context of suboptimal immune responses is an important driver of SARS-CoV-2 diversification. (Lines 88-91)"

R1.11: Line 112: E484V (rather than A484V?)- at least according to Figure 1.

Answer: We had denoted the change relative to the original BA.1 background that contains the E484A mutation. We have replaced this with E484V relative to the Wu-1 reference to match Figure 1.

R1.12: Line 114: 'Although viral isolation failed for specimens available from the index case..' I don't understand. Wasn't the virus isolated from P1?

Answer: We failed to isolate the virus from P1 specimens, but we were able to successfully culture viruses with the same mutation pattern from biospecimen collected from patients P2, P3 and P4. We have now further clarified this in the text by specifying the patient numbers (Lines 136-138, and Methods section, Lines 461-463). We believe that our inability to recover infectious viruses from the P1 specimen may be due to suboptimal storage of the clinical specimen for a prolonged time at 4C.

R1.13: Lines 116-117: Here, it seems that the forward transmissions were inferred based only on the common nonsynonymous substitutions found in spike. Why present this work this way, rather than by

presenting Figure 2A first (whole-genome analysis, and using both nonsynonymous and synonymous variation)? Please see my main comment about reorganization.

Answer: We considered all mutations when inferring transmission. As indicated in the original submission, all spike mutations (n=6) were nonsynonymous and there was one synonymous mutation within Orf1a (T6001C). We have rewritten the section on the forward transmission of the BA.1.23 lineage to better explain the rationale for limited local transmission based on the unique signature of 7 spike amino acid changes and the synonymous mutation at T6001C, which has only been detected in 6 cases globally, all of whom originated in NYC.

R1.14: Line 120: could you clarify how differing in age and gender from the 4 cases (P1-P4) provides an indication that there was independent but limited community spread of this subvariant?

Answer: Patients that were discharged from our hospital may also have been tested in the community. We therefore used the limited information on age and gender available for individuals tested by other laboratories in the NYC area to exclude the possibility that the two additional sequences reported by the CDC were obtained from our four patients P1, P2, P3 or P4. The reported age and sex of the CDC cases did not match our patients, suggesting that further, independent, spread occurred in the community. The age and gender information is provided in Table 1 and reference to this table has now been included in line 121-125.

R1.15: Mapping both synonymous and nonsynonymous substitutions on to the phylogeny (current Figure 2A; or a time-aligned version of this phylogeny) would be helpful

Answer: The signature nonsynonymous substitution (T6001C) has been added to the revised Figure 2A.

R1.16: Figure 2B: Are mSNV frequencies also potentially available for the GISAID sequences? (I.e., are there short read data available in the SRA that correspond to these GISAID consensus sequences?)

Answer: Unfortunately, GISAID does not support reporting of miSNV frequency data. However, as indicated in our response to comment R1.5, the raw sequencing data has now been made available through the SRA database, submission SUB12865927, under accession number SAMN33273632-SAMN33273655.

R1.17: Figure 2B: Rather than red and blue for in spike vs outside of spike, it would be more helpful to color by nonsynonymous vs synonymous (vertical lines could denote spike region)

Answer: We appreciate the suggestion. For simplicity, we have now marked nonsynonymous mutations occurring below the consensus level (<50%) with an asterisk. This is also specified in the figure legend.

R1.18: Instead of introducing the term 'mSNV', why not called it an 'iSNV' (e.g., McCrone et al. eLife)?

Answer: In our interpretation the term iSNV is used in McCrone et al. to refer to all intra-host variants (major and minor) present at a given position. We changed our term to miSNV, as an extension of iSNV that refers specifically to minor nucleotide variants.

R1.19: Line 185: how do we see this in Figure 2A?

Answer: The reviewer is correct that this statement requires the combined interpretation of panels 2A and 2C. We have now added references to both figure panels in the revised manuscript.

R1.20. In sum, this is an important case study that documents limited forward transmission of a highly divergent SARS-CoV-2 lineage that evolved in a chronically infected individual. The impact and interpretability of this work could be considerably improved from a restructuring of the manuscript. Beyond this, there are several other major comments (above) that, if addressed, would strengthen the manuscript.

Answer: We appreciate the reviewer's assessment that this is an important study. We thank them for their careful and constructive evaluations which have improved the manuscript's clarity. We are confident that we have successfully addressed their major comments.

Reviewer #2:

R2.1: Severely immunocompromised patients are at risk for severe and prolonged SARS-CoV-2 infection, and as a result, are an important potential source of viral mutation and development of variants. The authors characterize genetic mutations that occur over time, and attempt to demonstrate forward spread, which would be an important contribution to the field. However, there lacks data that clearly supports transmission of novel strains. Additionally, clinical details provided could be further clarified and link between antivirals and emergency of mutations should not be overstated.

Answer: The genotype of the BA.1.23 variant is unique and has been observed exclusively in New York City (NYC) based on a query of >13M global sequences deposited in GISAID. Moreover, the circulation of BA.1.23 was highly contained in time and space during the early diversification of lineage BA.1. We have further expanded on potential direct/indirect contacts between patients that are consistent with BA.1.23 transmission in the revised manuscript. We have also addressed the potential for recombination during the evolution of BA.1.23 (see also responses to comments R1.3 and R1.4 of Reviewer 1). Altogether, all available data strongly support a transmission chain from patient P1 to patients P2-P4 and two other community cases.

R2.2: Results, page 5: Authors characterize amino acid substitutions that developed in SARS-CoV-2 strain of patient 1. What is the baseline rate of amino acid substitutions of SARS-CoV-2 to help determine if persistent infection in an immunocompromised host is driver of evolution?

Answer: The revised manuscript now notes that the rate of amino acid substitutions varied during the course of infection in patient 1, as follows:

“The SARS-CoV-2 substitution rate varied throughout the course of infection; after an initial period of three weeks without changes in the consensus sequence, there was a rapid accumulation of substitutions between week 4 and week 12. On average, one substitution per week was observed during this period, corresponding to a rate of 52 substitutions/year – approximately two-fold higher than the global average of 26-27 substitutions/year.” (lines 106-111).

R2.3: Results, page 5: The authors reference isolating variants with shared amino acid substitutions, all of whom were heme malignancy patients. Was actual cluster analysis (time and location) performed to help confirm hospital transmission? If not, do not have enough data to state this is an outbreak.

Answer: In the revised manuscript, we have included additional metadata on potential direct or indirect contacts between patients P1-P4 (Lines 126-138). In summary, each patient overlapped with one or more other patients in the same unit at some point during their hospital admissions. Several of the patients with BA.1.23 stayed in rooms that were near each other, supporting a potential transmission chain from P1-P4 that is consistent with the pattern of emerging mutations in the BA.1.23 lineage. Please also see our answer to comment R1.3 by reviewer 1 for more details.

R2.4: Results, page 7-8: Patient received non-EUA-approved courses of therapies: e.g. 3-4 weeks of Paxlovid. What was the route of obtaining these therapies?

Answer: We do not know how our clinical colleagues decided on the antiviral treatments of their vulnerable patients.

R2.5: Results, page 8: Authors make multiple states about therapies received and subsequent detection of mutations or lack of persistent infection that may not be causative.

Answer: We appreciate the reviewers' point of view and agree that we cannot prove causation. We have carefully reviewed and adjusted the language describing the treatments received by each of the patients infected with BA.1.23 to avoid speculation.

R2.6: Discussion, page 11: Provide reference for low antibody levels being risk factor for immune escape mutations

Answer: The reference has been added to the revised manuscript [PMID: 33915337].

R2.7: Antivirals are were not studies looking at outcome of “eliminating persistent infection” but rather for reducing severity of infection. In Figure 3, it appears that cycle threshold of patients did increase (lower viral load) after receipt of antivirals?

Answer: The reviewer is correct. We currently have no specific antiviral treatment that has been tested in clinical trials to specifically eliminate persistent SARS-CoV-2 infection. We acknowledge that the benefits of antiviral treatment include reducing the severity of infection. As such, we have moderated our statement from “... failed to eliminate the persistent infection ...” to “... did not eliminate the persistent infection ...”. Line 299.

R2.8: Authors use the term “fully vaccinated” but do not provide details on which vaccines (we know that mRNA vaccines are more immunogenic than adenovirus vector, for example) and how many doses.

Answer: We have added the information on vaccine types to the revised manuscript, where available, as follows:

“Patient P2 had received four doses of Moderna mRNA vaccine, three at least four months prior to their first positive SARS-CoV-2 test, ...” (Line 204).

“Patient P3 was also fully vaccinated with two doses of Pfizer mRNA vaccine at least five months prior to their first positive test ...” (Line 210).

Reviewer #3:

Gonzalez-Reiche et al analysed SARS-CoV-2 persistent infections in immunocompromised patients. They described the emergence, transmission and subsequent evolution of the new Omicron sub lineage BA.1.23 in patients with persistent SARS-CoV-2 infection and replication. They observed that the initial substitutions were within the spike but continued replication led to substitutions in other viral proteins. The authors also showed that BA.1.23 variant was more resistant to neutralising antibodies induced post-booster vaccination or after BA.1 breakthrough infection compared to BA.1 and the ancestral Wuhan strain.

Understanding how SARS-CoV-2 variants emerge and further evolve in immunosuppressed hosts is crucial to develop strategies to treat these individuals in order to prevent the emergence of variants. This paper is highly relevant. The methodology looks appropriate to me. The paper is well written and the figures are very clear. I only have a few comments. Maybe a few additional points below-mentioned could be added to the discussion or clarified.

Answer: We appreciate the positive feedback on our manuscript. We addressed the comments and helpful suggestions to the best of our ability below and in the revised manuscript.

Major Comments:

R3.1: Line 113: when the authors speak about transmission, do they mean that these patients had some contact(s) at the hospital ? Are there any evidences they were in the same unit in the same time ?

Answer: We have added additional information to the revised manuscript to indicate that each of the patient infected with BA.1.23 had an overlapping stay in the same unit with at least one other patient, providing a potential route for transmission through direct or indirect contacts. Please also see also our answers to comment R1.3 by Reviewer 1 and R2.1 by Reviewer 2.

R3.2: Line 117: do the authors mean B-cell and T-cell deficiencies in all patients ? Or only B-cell deficiencies ?

Answer: We do not have further diagnostic information on the patients, other than that they had underlying hematologic malignancies.

R3.3: Lines 181-182: did the authors try to discriminate vaccine-induced and infection-induced antibodies by measuring anti-N ? Even though I presume IVIg may also bind to N. More globally, I was wondering whether the authors compared both anti-S and anti-N in comparison to see if residual antibodies of different specificities were measured.

Answer: The SARS-CoV-2 antibody test used in the clinic measures spike binding IgG antibodies. The test, thus, does not allow us to distinguish between infection- or vaccine-induced responses. Future studies investigating persistent infections in real time are needed to address whether antibodies of different specificities are present in these persistent infections.

R3.4: In addition, I don't see Figure 2C mentioned in the text. It is probably 3B.

Answer: We have corrected the figure reference in the revised manuscript.

R3.5: Line 198 and line 205: the data suggest that a suboptimal level of antibodies may lead to a selective pressure. Do the authors mean these antibodies were not functional (neutralising, Fc receptor function) ? I was wondering whether it was only related to the magnitude or also a lack of functionality. In addition, related to the use of MAb treatment, is there a way to play with the dose and/or length of treatment to make sure the benefit outweighs the risk of selective pressure?

Answer: These are excellent questions that we cannot answer with our current dataset. Future studies investigating persistent infections in real time are needed to address whether antibodies of different specificities are present in these persistent infections.

R3.6: Did the authors analyse if currently available MAb-based therapies could neutralise BA.1.23 ?

Answer: We appreciate the suggestion. We have not yet tested to what extent BA.1.23 variants are resistant to monoclonal antibodies, but we are in the process of characterizing the different major and minor BA.1.23 variants that emerged over the eight months of the study. These data will be presented in a follow up manuscript.

R3.7: Line 262: Is there any role of T cells in the intrahost evolution ? Can selection be also driven by T cell escape ? Or is it only related to antibody response ? Do the authors have any T cell data in these patients ?

Answer: This is an interesting suggestion. Unfortunately, peripheral blood mononuclear cells were not available from these cases and cellular responses could not be determined. The serological data presented in this manuscript is restricted to available clinical measurements or testing of residual sera from diagnostic specimens. Future studies aimed at following persistent infections in real time will include assessments of T cell responses in addition to measuring humoral responses.

R3.8: Are there any forms of immunosuppression which are more susceptible to lead to the emergence of variants and mutations leading to escape patterns ?

Answer: These are very important questions that require larger datasets. The four cases included in this study are not a large enough sample size to provide robust answers. We hope that the current publication will start potential collaborations to pool persistent cases to obtain large enough sample sizes to address these critical questions.

Minor comments:

R3.9: Figure 3: legends B and C were inverted

Answer: Thank you for pointing out this mistake. We have corrected the error.

R3.10: Line 383: typo “different time points”

Answer: Thank you for pointing out this typo. We have made the correction in the revised manuscript.

R3.11: Extended data Figure 1: what does n mean ?

Answer: The notation “n” represents ambiguous bases with coverage below the base-calling threshold of 10X. This has now been clarified in the figure legend.

R3.12: Extended data Figures 2 and 3: could the authors use the same colors for New and BA.1.23 in each figure (Extended Figures 2, 3 and 4)?

Answer: We are now using the same colors across the extended figures.

Reviewer #4:

Major comments

R4.1: The epidemiological contexts linking the index case to the other individuals should be stated. In the relevant Results sub-section (lines 108- 128), the authors describe forward transmission from the index case (P1) to five other individuals, comprising three immunocompromised patients (P2 P3, P4) and two community members (GISAID S1 and S2), based on the shared presence of a unique combination of 7 spike protein mutations.

Is there available information on how the other individuals came into contact with the index case, such as whether the three patients shared the same ward? Did the two additional individuals from the NYC area come into contact with the index case or is there a possibility they were contaminated elsewhere? This would help the reader understand the transmissibility of the BA.1.23 variant.

Answer: We have added additional information to the revised manuscript to indicate that each patient had an overlapping stay in the same unit with at least one other patient, providing a potential route for transmission through direct or indirect contacts. We do not have information on the two additional cases from NYC apart from their age and gender. We speculate that these cases either had direct/indirect contact with P2 after this patient was discharged, or that the two cases had unknown contacts with our health system. Please also see also our answers to comment R1.3 by Reviewer 1 and R2.1 by Reviewer 2.

R4.2: Line 214-215 Why were these 7 mutations selected to be the defining mutations of BA.1.23 out of all detected mutations, not including minority variants (Figure 1)?

Answer: The lineage designation was made based on a set of signature mutations (7 non-synonymous mutations in spike and 1 synonymous mutation in Orf1a) that were shared across all cases of the transmitted BA.1.23 variant. Minority variants were present mostly in a subset of samples from one of the transmission cases.

R4.3: Line 203-204. P4 had low levels of antibody titres at the time of the first positive PCR although they were unvaccinated. Could this be an indication of a previous Covid infection? Are past infection histories of the other cases known?

Answer: Past infection histories for P1-P4 are limited to the test data recorded in the electronic medical records, which revealed no evidence of prior SARS-CoV-2 infections. P4 has had frequent interactions with our health system since the start of the pandemic in February of 2020. During this time the patient was tested on average every 1-2 months and there were no positive tests prior to day 104 in our study. The consistent negative testing history suggests that there was no prior infection before BA.1.23, although we cannot exclude the possibility that P4 (or other patients) tested positive elsewhere.

Minor comments

R4.3: Line 45-46. I find this sentence to be misleading as it sets up the reader to imagine forward

transmissions from three index cases. While it is true the study includes three individuals with persistent infections, the fact that there was only one source of forward transmission affects how the transmissibility of the emerging variant BA.1.23 is perceived.

Answer: In the revised manuscript we provide additional information on potential direct/indirect contacts between patients to provide further context on potential transmission chains (Lines 126-138). Please also see response to question R1.3.

R4.4: Figure 1. The legend should explain why some mutations are in bold.

Answer: An explanation has been added to the legend in the revised manuscript.

R4.5: Figure 3C and 3B legends should be swapped.

Answer: We apologize for the oversight and have made this change in the revised manuscript.

R4.6: Line 160 "Figure. B2".

Answer: This has been corrected in the revised manuscript.

R4.7: Line 383 "time pointes".

Answer: This typo has been corrected in the revised manuscript.

R4.8: Figure 4 Neutralisation titre panels. Presently, the GMT values are too easily confused as X-axis values. It might clarify the graph to remove the X-axis ticks and place the GMT values in a table to dissociate them from the X-axis.

Answer: We thank the reviewer for pointing out this possible source of confusion. We have revised the figure accordingly.

R4.9: Extended Data Figure 2: Do the authors mean diamonds instead of triangles?

Answer: This has been corrected in the revised manuscript.

Reviewers' Comments:

Reviewer #1:

Remarks to the Author:

The authors have addressed all of my concerns, and I believe the findings in this manuscript are an important contribution to our understanding of whether prolonged infections are able to onward transmit.

Reviewer #2:

Remarks to the Author:

The authors have made additions and changes to languages that appropriately address comments.

Reviewer #3:

Remarks to the Author:

The manuscript has been significantly improved based on reviewers' comments. I am happy with the replies to my comments.

Reviewer #4:

Remarks to the Author:

Gonzalez-Reiche et al. followed the rise and onward transmission of the BA.1.23 variant from one index case to five individuals, where isolates from all six individuals showed a unique set of seven spike protein mutations. The index case and two patients of forward transmission were immunocompromised and the authors posit that these conditions are associated with expanded intrahost mutations. The authors also demonstrated that the new BA.1.23 variant is more resistant to antibodies induced by booster vaccination or by BA.1 breakthrough infection.

This work is highly relevant as it provides evidence that persistent infections in immunocompromised hosts can lead to transmissible variants with potentially higher virulence.

The lack of epidemiological context regarding contact between the study individuals has been addressed in this latest iteration of the manuscript. Other comments have been also addressed satisfactorily.

I have no further comments.